# Analysis of aerosol-cloud interactions and their implications for precipitation formation using aircraft observations over the United Arab Emirates

Youssef Wehbe[1], Sarah A. Tessendorf[2], Courtney Weeks[2], Roelof Bruintjes[2], Lulin Xue[2], Roy Rasmussen[2], Paul Lawson[3], Sarah Woods[3], and Marouane Temimi[4]

[1]National Center of Meteorology, Abu Dhabi 4815, UAE
[2]Research Applications Laboratory, National Center for Atmospheric Research, Boulder, CO 80307, USA
[3]Stratton Park Engineering Company, Boulder, CO 80301, USA
[4]Department of Civil, Environmental and Ocean Engineering, Stevens Institute of Technology, Hoboken, NJ 07030, USA

*Correspondence to*: Youssef Wehbe (ywehbe@ncms.ae)

**Abstract.** Aerosol and cloud microphysical measurements were collected by a research aircraft during August 2019 over the United Arab Emirates (UAE). The majority of science flights targeted summertime convection along the eastern Hajar mountains bordering Oman, while one flight sampled non-orographic clouds over the western UAE near the Saudi Arabian border. In this work, we study the evolution of growing cloud turrets from cloud base (9 °C) up to the capping inversion level (-12 °C) using coincident cloud particle imagery and particle size distributions from cloud cores under different forcing. Results demonstrate the active role of background dust and pollution as cloud condensation nuclei (CCN) with the onset of their deliquescence in the sub-cloud region. Sub-cloud aerosol sizes are shown to extend from submicron to 100 µm sizes, with higher concentrations of ultra-giant CCN (d>10 µm) from local sources closer to the Saudi border, compared to the eastern orographic region where smaller size CCN are observed. Despite the presence of ultra-giant CCN from dust and pollution in both regions, an active collision-coalescence (C-C) process is not observed within the limited depths of warm cloud (<1000 m). The state-of-the-art observations presented in this paper can be used to initialize modelling case studies to study the influence of aerosols on cloud and precipitation processes in the region and to better understand the impacts of hygroscopic cloud-seeding on these clouds.

## 1 Introduction

Aerosol particles are key components of the atmosphere and have multi-scale impacts on Earth's climate and hydrological cycle, primarily through radiative transfer and precipitation formation. Aerosol effects are grouped into direct radiative effects (Ming et al., 2005) and indirect aerosol-cloud interactions (Lohmann et al., 2010). The magnitude of uncertainty from indirect effects remains more difficult to quantify compared to that from direct effects (Solomon et al., 2007). This is primarily due to the complexity of the microphysical processes involved and their interdependencies (Morrison et al., 2005). Furthermore, the

contributions from thermodynamic and dynamic factors introduce additional complexities to the precipitation generation process. Hence, assessing the impact of aerosols on cloud microphysics and precipitation generation has been a longstanding research area with many processes yet to be fully understood (Morrison et al., 2020;Flossmann et al., 2019;Solomon et al., 2007;Squires, 1958;Rosenfeld, 1999). Our current knowledge is based primarily on observing the influence of background aerosols on the number and size distribution of hydrometeors (Rosenfeld et al., 2008;Freud et al., 2008;Tao et al., 2007;Rosenfeld, 2000;Feingold et al., 1999). Isolating these impacts has been shown to be particularly challenging in polluted and dusty environments where the physiochemical properties of aerosols are continuously altered between large dust particles, fine particle pollution as well as complex aggregates depending on both regional and local forcing (Abuelgasim and Farahat, 2020;Filioglou et al., 2020;Semeniuk et al., 2014).

The role of dust aerosols as ice nucleating particles (INPs) is well established in the literature (Kanji et al., 2019;Atkinson et al., 2013;Hoose and Möhler, 2012;Prenni et al., 2009;Möhler et al., 2006). Dust INPs have been shown to prolong cloud lifetimes and enhance precipitation in relatively unpolluted continental-tropical and maritime regimes (Liu et al., 2020;Koren et al., 2005). Alternatively, when ingested into strong convective updrafts, large populations of mineral dust INPs tend to limit the growth of hydrometeors in the upper levels due to their competing effect for available water vapor (Kant et al., 2019;Min et al., 2009;DeMott et al., 2003;Sassen et al., 2003). On the other hand, the role of dust as cloud condensation nuclei (CCN) in polluted desert environments with diverse aerosol physio-chemistry is less understood. The recent study by Chen et al. (2019) showed non-linear responses of precipitation to dust when it serves as both CCN and INP.

Dust introduces large particles with diameters exceeding 2 µm which could serve as giant CCN (GCCN) (Jensen and Nugent, 2017;Yin et al., 2000;Jensen and Lee, 2008). This alludes to more effective droplet collection during the collision-coalescence (C-C) process. However, Rosenfeld et al. (2001) attributes the reduction in cloud droplet effective radii ($r_e < 14$ µm) over the Saharan desert to the presence of large concentrations of submicron CCN originating from desert mineral dust. This is shown to inhibit C-C and warm rain formation, exacerbating a reduction in precipitation over the Sahara region. A similar conclusion is derived from the modelling studies of Flossmann and Wobrock (2010). These findings contradict with previous work suggesting enhanced C-C by GCCN regardless of the fine-mode concentrations (Johnson, 1982;Takeda and Kuba, 1982). Aside from size and concentration, aerosol chemistry plays an important role in determining the nucleation properties of dust particles. Laboratory deliquescence experiments show that pure insoluble mineral dust particles remain hydrophobic at 100% relative humidity (RH), while the aggregation of dust particles with soluble compounds (e.g. NaCl) results in their deliquesce at 74% RH (Wise et al., 2007). Similarly, compounds from anthropogenic pollution (sulfates) interact with mineral dust to produce internally-mixed aggregates which deliquesce at 80% RH (Semeniuk et al., 2015).

Located along the central portion of the dust belt, the Arabian Peninsula is largely considered a hotspot for atmospheric dust emissions (Karagulian et al., 2019;Beegum et al., 2018;Hofer et al., 2017). The UAE is situated on the southeastern coast of the Arabian Peninsula and records an annual rainfall average less than 100 mm, which is representative of the region's scarce

rainfall amounts (Wehbe et al., 2018). Aiming to augment natural rainfall over the country, the UAE has implemented an operational cloud-seeding program for the past two decades – the longest ongoing program in the region (Almazroui and

65 Farrah, 2017). In addition to an in-depth understanding of the local meteorology, cloud-seeding programs require an accurate characterization of the background aerosol particles and their potential to serve as CCN or INPs. To this end, the UAE Unified Aerosol Experiment (UAE[2]) was the first comprehensive airborne assessment of aerosol characteristics in the Arabian Gulf (Reid et al., 2006). The measurements indicated an abundance of sulfate-dominant fine-mode aerosols which may have strong influences for cloud and precipitation formation. Further evaluation of the impact of background aerosols as CCN/INP on

cloud microphysics and precipitation was recommended as a high priority to refine operational cloud-seeding activities over the UAE (Semeniuk et al., 2014;Breed et al., 2007).

In line with the recommendations from the UAE[2] work, a flight campaign was conducted in August 2019 by the Stratton Park Engineering Company (SPEC) Inc., supported by the UAE Rainfall Enhancement Program (UAEREP). The aircraft was equipped with state-of-the-art cloud physics instrumentation, outlined in Lawson et al. (2019), to sample summertime

convective clouds over the UAE (see Sect. 4). Here, we use in situ aerosol and cloud microphysical measurements from the SPEC UAE campaign to assess the role of background aerosols on natural precipitation formation. Representative measurements from two separate flight cases (August 12[th] and 19[th], 2019) are used to study dominant eastern orographic convection along the Hajar mountains (Branch et al., 2020) and the less frequent southwestern convection associated with the Arabian heat low (AHL) near the Saudi Arabian border (Steinhoff et al., 2018).

## 2 Regional setting

Given its geographical setting, the UAE is impacted by both mesoscale sea-land breezes and microscale orographic storms with lifetimes less than 30 minutes. Interestingly, despite its minimal contribution to the country's mean annual rainfall, the summer season records rainfall accumulations exceeding 100 mm over the eastern Hajar mountains bordering Oman (Wehbe et al., 2020;Wehbe et al., 2017). In a recent study, Branch et al. (2020) calculated back-trajectories of summertime convective

events over the eastern UAE by applying thermal radiance thresholds to the European Organization for the Exploitation of Meteorological Satellites (EUMETSAT) Meteosat-7 and 10 retrievals during a 7-year period. While convection was found to be predominantly fostered along the Hajar mountain peaks (>80% of events) around noon local time, a diurnal propagation of this convection outflows to the west during the afternoon periods – coincident with the easterly sea breeze (Remiszewska et al., 2007).

On the other hand, rare non-orographic summertime precipitation has also been reported over the UAE. Steinhoff et al. (2018) linked such events over the southwest to increased convection and latent heat over the Arabian Sea associated with the active phases of the southwest Asian monsoon. This was shown to strengthen the circulation of the AHL, which shifted moisture transport in favor of deep convection initiation over the southwest desert inland. Kumar and Suzuki (2019) also used Meteosat-

10 retrievals for a satellite-based assessment of cloud climatology and seasonal variability of microphysical properties over

the UAE. They reported a high occurrence of mixed-phase convective clouds along the coast and northeastern mountains during summer periods. Their analyses of cloud types showed higher warm cloud fractions over the mountainous regions, while cold cloud fractions were higher over the Arabian Gulf during winter and strictly localized along the Hajar mountains in summer. This is consistent with the radar-based analysis of Breed et al. (2007) which showed the highest frequency of mixed-phase clouds from orographic convection along the Hajar mountains during the summer months. Consequently, the

August 2019 period was selected as the target period for the SPEC UAE campaign. A total of 11 science flights (SFs) sampled frequent orographic clouds west of the Hajar foothills, in addition to one set of non-orographic clouds over the southwest near the Saudi border. Figure 1 shows the flight tracks during the August 12th and 19th cases, referred to hereafter as SF1 and SF4, respectively, and the composite reflectivity from the UAE National Center of Meteorology (NCM) radar network.

## 3 Synoptic situation and thermodynamic profiles

Figure 2a-d shows the synoptic situation using the European Centre for Medium Range Weather Forecasts Reanalysis 5 (ERA5) product (Hersbach and Dee, 2016) within 1-hour before each flight – at 11:00 and 12:00 UTC for SF1 and SF4, respectively. The mean sea-level pressure (MSLP) during SF1 (Fig. 2a) shows the two typical pressure lows impacting the UAE during the summer season, namely, the Pakistan-India low (Bollasina and Nigam, 2011) and the Arabian Gulf low (Bitan and Sa'Aroni, 1992). However, the SF4 case (Fig. 2b) occurred during the less-frequent dominance of the AHL (Steinhoff et

al., 2018) over the southwestern region. The low-level atmospheric thickness (LLAT) index is used for heat low detection (Fig. 2c,d) as proposed by Lavaysse et al. (2009) for the West African heat low. The varying threshold (red color scale) is defined as the 90th percentile of the LLAT cumulative probability distribution function (i.e. the highest 10% values of LLAT) computed from all the grid points (hourly scale used here). The active AHL extends from central Saudi Arabia to the southwest of the UAE during SF4 (Fig. 2d).

To further assess discrepancies between the two flight cases, air mass back-trajectories were computed using the Hybrid Single Particle Lagrangian Integrated Trajectory (HYSPLIT) model (Stein et al., 2015). Figure 2e,f show the 1.5 and 4.5 km level back-trajectories to two locations within the flight area and a third control location to the south for each flight date to distinguish between regional and local air mass trajectories. During SF1, all low-level (1.5 km) trajectories (Fig. 2e) show the dominant sea breeze convergence inland and toward the Hajar foothills, while the upper-levels (4.5 km) indicate long-range easterly

transport from the Indian regime. However, during SF4, more locally driven trajectories aligning with the extent of the AHL are observed over the Empty Quarter desert area to the west.

Figure 3 shows the Skew-T profiles comprised of the aircraft-observed temperature and dew point temperature from the surface at 970 hPa (43 °C) to 400 hPa (-15 °C) for both flight case. Co-located ERA5 profiles (Fig. 3a,b) and radiosonde observations (Fig. 3c,d) from Abu Dhabi (AD) Airport (12:00 UTC) are overlaid for each case. Despite the different synoptic situations

during the SF1 and SF4 cases, similar cloud base heights and temperatures of 3.5 km and 9 °C, respectively, are observed. These are typical values which fall within the long-term interquartile range of summertime convection (35-year analysis of AD Airport soundings, not shown). However, much drier conditions are evident in SF4 compared to SF1 soundings above the 700-hPa level. Although the ERA5 dataset provides a high vertical resolution (~250 m), it represents the mean conditions within a single ERA5 grid cell (~30 km). Hence, the mean condition as presented in the EAR5 data may not represent the aircraft observed instance due to the high variability of the low- to mid-level atmospheric conditions in this region. Strong upper-level temperature inversions at around -5 °C and -3 °C are observed during SF1 and SF4, respectively. The large differences in the low-level dew point profiles between the aircraft and AD soundings is primarily due to the coastal location of the latter. The noise in the dew point temperature profiles between the 700 and 500 hPa levels indicates the aircrafts' motion in and out of clouds.

## 4 Dataset and methods

### 4.1 Aircraft instrumentation

The science flights were conducted by the SPEC Learjet 35A. The capability of the Learjet to conduct rapid maneuvers and swift ascents/descents was key for sampling the short-lived UAE clouds while complying with local Air Traffic Control (ATC) restrictions. The aircraft was equipped with state-of-the-art cloud physics instrumentation used on multiple platforms throughout various airborne campaigns (Lawson et al., 2019). The instruments deployed for the UAE campaign included a Condensation Particle Counter (CPC), Passive Cavity Aerosol Spectrometer Probe (PCASP), multiple optical array probes (OAPs) and scattering probes (see Table 1). Figure 4 shows the Learjet in-flight over Al Ain and selected instruments.

The scattering probes included a standalone Fast Cloud Droplet Probe (FCDP) (Lawson et al., 2017) and Fast Forward-scattering Spectrometer (FFSSP), with the latter upgraded to reduce the effects of ice shattering and processing lag time (O'Connor, 2008). The scattering probes' retrieval and processing mechanisms are provided in Knollenberg (1981) and in the Appendix of Lawson et al. (2017). The instrument suite also included the Hawkeye (Woods et al., 2018), a recently developed composite system housing its own FCDP, two-dimensional stereo (2D-S) probe and Cloud Particle Imager (CPI) (Lawson et al., 2001). The CPI provides high resolution (2.3 µm) digital images of cloud particles for ice-phase habit identification (Woods et al., 2018). The triple-coincident observations from the Hawkeye were especially useful for the mixed-phase clouds targeted in the UAE, where a size threshold was applied on the 2D-S as a trigger for the CPI to capture ice particle habits within a large population of cloud drops. Only the CPI component of the Hawkeye was used in this paper, given the Hawkeye-FCDP measurements required further calibration and testing.

The OAPs included a total of three 2D-S probes (two standalone and one within the aforementioned Hawkeye) and one High Volume Precipitation Spectrometer (HVPS) described in Lawson et al. (2006) and Lawson et al. (1993), respectively. A total

of five channels (three vertical and two horizontal) provided redundant measurements at 10 µm resolution (see Table 1) which served for data quality control (see Table 1). Post-processing of the OAP imagery involved noise filtering and corrections for particle overlapping and ice shattering as outlined in Lawson (2011). Size estimates from the two image processing methods $M_4$ and $M_7$ described in Lawson et al. (2017) were used for smaller (round drops) and larger (irregular ice) shapes, respectively.

As listed in Table 1, cloud droplet size distributions were derived using the well-established FFSSP and 2D-S instruments, while the usage of the standalone FCDP was limited to coarse mode aerosol measurements to complement the PCASP accumulation mode measurements. The fine-mode PCASP measurements were quality controlled by adjusting the sample volume to account for pressure-induced pump rate lags, especially at altitudes exceeding 4.5 km. The first two size bins (0.1– 0.12 µm) were also excluded from the total concentration calculations to avoid data contamination from excessive noise. All PCASP total concentrations were filtered for out-of-cloud measurements by applying dual thresholds on the FFSSP total number concentration (20 cm$^{-3}$) and hot-wire liquid water content (LWC; 0.01 g.m$^{-3}$), above which is considered a cloud pass (Korolev and Isaac, 2006). Similar to a CCN counter, but forced with higher supersaturation, the CPC provided counts of ultra-fine mode aerosols (0.01–3 µm) through deliberate water-based condensation of intercepted particles to reach sizes detectable by a laser counter (Wang et al., 2015;Liu et al., 2006). Finally, an Aventech Model Aircraft Integrated Meteorological Measurement System (AIMMS) logged 1-Hz basic meteorological variables and air motion measurements (Beswick et al., 2008), while a Nevzorov hot-wire probe (Korolev et al., 1998) measured the total water content and LWC – used here as the reference value compared to LWCs registered by the scattering probes and OAPs.

## 4.2 Cloud penetration selection and classification

A total of 13 and 22 cloud penetrations were conducted during SF1 and SF4, respectively. All flight ascents were southward out of Al Ain, climbing up to 7 km (flight altitude limit set by ATC) and penetrated clouds on the descent back over Al Ain (SF1) or the southwest (SF4). Growing turrets were penetrated nearest to their tops to document initial ice formation (i.e. first ice). However, as a result of the 7 km ATC limit, there are no cloud measurements at levels colder than -12°C. Penetrations were also made around 300 m above cloud bases to measure the microphysical and dynamic properties of the updrafts. Here, the analysis is primarily focused on penetrations from growing turrets, aiming to study early-stage cloud conditions and their evolution. Penetrations from precipitating, decaying or dissipating clouds are excluded from the analysis (used only for demonstration).

To minimize data contamination from entrainment and evaporation effects or artificial spikes in the 1-Hz acquisitions, only measurements from cloud cores are extracted. Avoiding entrainment effects is particularly challenging in this region's dry environment where even the most undiluted penetrations are contaminated by downdrafts. The analysed cloud cores were sub-adiabatic in all cases. The Steady State (SS) and Maximum Drop Concentration (MDC) methods proposed by Tessendorf et al. (2013) are used to manually identify the most isolated cloud core measurements within each penetration time series. Both

methods are used to identify the undiluted core and better attribute the measured spectra to the sub-cloud aerosol. As the name suggests, the SS method is based on the assumption that LWC and droplet concentrations remain relatively constant during a 3-second window, while the MDC method averages the measurements over the 3-second window of maximum concentrations to better correlate with sub-cloud aerosol measurements. Both methods are applied by manual inspection of the time series data. We first present the background aerosol analysis, followed by a detailed microphysical analysis and inter-comparison between the two.

## 5 Results and discussion

### 5.1 Aerosol measurements

Figure 5 shows the 3D flight tracks from both cases and their PCASP total concentration measurements throughout. For both flight cases, the PCASP concentrations show a 10-fold increase on the descent over Al Ain compared to those from the ascent, even at low altitudes (below 1500 m). This can be explained by multiple co-occurring effects. The convective outflow from the Hajar mountains toward Al Ain and the coincident timing of the SF1 descent with the peak in diurnal sea breeze activity (around 12:00 UTC) (Eck et al., 2008) can form a convergence zone over the region of Al Ain which is subject to orographic trapping of dust along the Hajar foothills (Schwitalla et al., 2020). The effect of the orographic dust trapping is evident from the higher PCASP descent concentrations from SF1, which is further east toward the Hajar ridge, compared to that of SF4 with a direct descent over Al Ain (see Fig. 1). Also, the ascent-descent magnitude split in the PCASP profiles are less pronounced for flights on non-cloud (dry) days (i.e. SF5 and SF9 – not shown here). This indicates the strong contribution from the outflow of convection and thunderstorms during cloudy days (SF1 and SF4) coupled with the prevalent dust-laden haboob winds (Miller et al., 2008). The range of penetrated cloud temperatures are shown for each flight, with high concentrations of aerosols (~1000 cm$^{-3}$) at cloud base (~9 °C) in both cases.

The vertical profiles of PCASP total number concentrations are shown in Figure 6. During the SF1 ascent, the concentrations decrease from around 400 cm$^{-3}$ to 100 cm$^{-3}$ within the boundary layer up to 2 km, and increase again to 180 cm$^{-3}$ around 4 km. Additional variations in the change in aerosol concentration with altitude are observed up to the top 7 km level, while the descent shows relatively less variability. This suggests the presence of multiple dust and aerosol layers (labelled L1-4 on Fig. 6a) on the cloud-free ascent and more mixed conditions with higher concentrations are observed on the descent with cloud development. A well-mixed profile is observed on both the ascents and descents of SF4 as clouds were present over the southwest and Al Ain, but the magnitude split is still evident between the two profiles. The observed stratification of dust and aerosol layers is in line with the ground-based Lidar observations from Filioglou et al. (2020) over Al Dhaid – situated along the Hajar foothills at 130 km from Al Ain – where they report four separate layers up to 6 km during August 2018. The stratification is imposed by gravitational waves produced from the sea breeze-mountain overpasses during the afternoon period

and by multiple temperature inversions (see Fig. 3) which are frequently observed during the summer months (Weston et al., 2020).

The time series of altitude (and temperature) measurements from SF1 and SF4 are shown in Figures 7a and 7c, respectively. Cloud bases are sampled at around 3 km and 9 °C in both cases. The selected time series of the sub-cloud and in-cloud LWC
(hot-wire), PCASP and FFSSP total number concentrations are displayed in Figures 7b and 7d. Coincident peaks in the LWC and FFSSP total concentrations indicate a cloud base penetration, while intermediate intervals are considered sub-cloud measurements. In both cases, the mean sub-cloud PCASP total number concentrations vary around 500 cm$^{-3}$, but with higher variations (up to 1000 cm$^{-3}$) during SF1. The mean background total concentration of 500 cm$^{-3}$ is in close agreement with those recorded over the Sahara desert by Rosenfeld et al. (2001). On the other hand, the LWC shows excessive noise during the SF4
sub-cloud passes with peaks of around 0.2 g.m$^{-3}$ compared to SF1 (<0.05 g.m$^{-3}$), which may be attributed to the hot-wire response to heavy dust (and haze) loading from local pollution over the southwest. This is corroborated by the more frequent FFSSP measurements during SF4 compared to SF1, suggesting the presence of large background aerosols from local pollution and dust.

The FFSSP concentrations are always less than PCASP concentrations during the SF1 sub-cloud time series (see Figure 7b).
However, in the case of the dustier sub-cloud conditions of SF4 (Figure 7d), there are instances where the peak FFSSP concentrations are larger than the PCASP (and CPC) concentrations, with relative differences less than 20%. Differences in flow rate, refractive index and relative humidity-dependent errors introduce inconsistencies in the calibration curves of the optical sizing instruments with an average uncertainty of 28% considered acceptable for the inter-comparison of their measurements (Moore et al., 2004;Reid et al., 2003).

The CPC generally records less variability as it samples aerosols at a significantly larger volumetric rate (50 cm$^3$/s) compared to that of the PCASP (1 cm$^3$/s) (Cai et al., 2013;Wiedensohler et al., 2012). Figure 7d shows larger concentrations of PCASP compared to CPC particles during the initial interval (13:44–13:45:30), and vice versa for the final interval (13:54–13:57). However, the CPC particle concentrations fall within the 20% uncertainty margin (Rosenberg et al., 2012) of the PCASP particle concentrations during the inner interval (13:47–13:54). The comparable measurements during the majority of the SF4
flight track over the southwest (13:47–13:54) suggests a smaller concentration of ultra-fine (0.01–0.1 um) compared to larger particles (0.1–3 µm).

Figure 8 shows the background aerosol size spectra from SF1 and SF4 merged from overlaps between the PCASP, FCDP, FFSSP and 2D-S10 measurements. A higher concentration of fine-mode aerosols between 0.1–0.2 µm is observed during SF1 compared to SF4 with a peak difference of approximately 5×10$^5$ L$^{-1}$ µm$^{-1}$. Alternatively, higher tail concentrations from ultra-
245 giant sizes of 20–50 µm are recorded during SF4 compared to SF1. This is in line with the previous suggestion of higher ultra-

giant CCN loading from local pollution and dust aggregation over the southwest compared to the eastern regime (Semeniuk et al., 2015).

## 5.2 Cloud microphysics

The SS and MDC methods gave matching instances for the most undiluted cloud core measurements (closest to adiabatic cores) with adiabatic fractions between 0.6–0.8. Figure 9 shows the coincident imagery from the 2D-S10, Hawkeye-CPI and cockpit camera at multiple levels of growing turrets sampled during SF1. The altitude, temperature, vertical velocity range, hot-wire LWC, FFSSP total number concentration and median volume diameter (MVD) are listed in Table 2 for each level. The FFSSP and FCDP (not listed) concentrations just above cloud base (9.1 °C) are comparable at 800 and 1144 cm$^{-3}$, respectively. The maximum LWC of 0.8 g.m$^{-3}$ is coincident with the peak updraft velocity of 3.1 m.s$^{-1}$ and an 8.7 µm MVD. At the next level (4.5 °C), slightly lower FFSSP (621 cm$^{-3}$) and higher FCDP (1598 cm$^{-3}$) concentrations are recorded with a stronger updraft (6.9 m.s$^{-1}$) and higher LWC (1.2 g.m$^{-3}$) compared to the previous level. The marked difference between the FFSSP and FCDP concentrations at this level suggests that more than 50% of the intercepted cloud droplets and large aerosols are within the medium size range (d<6 µm) which are better captured by the FCDP. Similar values are also observed at the 3.2 °C level with a comparable updraft (6.2 m.s$^{-1}$) and LWC (1.2 g.m$^{-3}$). The 2D-S10 and CPI imagery for the lowest three levels (Fig. 9c–e) confirm the dominance of small drops (d<10 µm) with the exception of a couple of large drops captured by the 2D-S10. This is consistent with the calculated MVD values ranging from 8.9 to 10.9 µm from the three warm levels.

In terms of ice processes in the upper portion, ice is observed by the CPI and 2D-S10 at the -12.4 °C level. Very few ice particles show a habit of sector plates, as expected by nucleation at -12 °C with relatively high liquid water contents of 1.1 g.m$^{-3}$ (Pruppacher and Klett, 1980). This observation is within a decaying turret (-12.4 m.s$^{-1}$ peak downdraft). Interestingly, at the highest sampled level (-12.6 °C) of a growing pileus cloud, a dominant population of liquid drops (d<50 µm) are observed in the absence of ice particles with a LWC of 1.4 g.m$^{-3}$ and a strong 17.8 m.s$^{-1}$ updraft. The calculated MVDs remain less than 20 µm in both of these sub-freezing temperature penetrations.

Following the same format of Figure 9, the growing cloud turret penetrations at multiple levels during SF4 are shown in Figure 10. At cloud base (8.6°C), comparable concentrations from the FFSSP (541 cm$^{-3}$) and FCDP (605 cm$^{-3}$) are recorded at the peak 3.2 m.s$^{-1}$ updraft and 0.2 g.m$^{-3}$ LWC. The CPI and 2D-S10 capture a few 100 µm drops which appear to be deliquesced dust and pollution aggregates, as suggested from Figure 7d. At the next slightly colder cloud base (8.3°C), more 100 µm drops are captured by the CPI with higher total concentration and LWC of 0.5 g.m$^{-3}$. However, at the next -0.3 °C and -3.5°C levels, small drops (d<50 µm) dominate the imagery similar to SF1. Additionally, the LWC remains high (1.2–1.3 g.m$^{-3}$) for the upper sub-freezing levels where graupel and ice irregulars (0.5–1 mm) are detected at -10.6 and -11.2 °C. The strong updraft (24.4 m.s$^{-1}$) may have carried a limited number of large particles aloft to serve as ice nuclei at the -11.2 °C level. Hence, the larger drops observed at the lower two levels (8.6 and 8.3°C) most likely originate from melted ice that has fallen from above.

To inter-compare the evolution of droplet sizes with altitude, Figure 11 shows the merged size spectra from the PCASP, FFSSP, 2D-S50 and HVPS measurements during SF1 (a) and SF4 (b) at the levels presented in Figures 9 and 10, respectively. For the 9.1°C level of SF1, a peak concentration of $10^5$ L$^{-1}$ µm$^{-1}$ is shown for the sizes between 6–10 µm, which is consistent with the imagery-based analysis (Fig. 9e). For sizes larger than 10 µm, the concentrations drastically decrease to around 10 L$^{-1}$ µm$^{-1}$, while the 2D-S50 shows sizes up to 200 µm sizes at minimal concentrations of $10^{-3}$ L$^{-1}$ µm$^{-1}$. At the next two 4.5 °C and 3.2°C levels, no broadening is observed with a similar decrease in concentrations for sizes larger than 10 µm. Similarly, the cloud base penetration of SF4 at 8.6 °C reveals a consistent peak in the concentrations of 6–10 µm sizes, followed by a sharp decrease in concentrations thereafter. However, the 8.3 °C and -0.3 °C levels show slight broadening with the peaks extending to 20 µm, which may be explained by the hygroscopic characteristics of the southwest ultra-fine background aerosols (see Fig. 7d). However, the presence of large haze particles, turbulence and downdrafts may have also contributed to the larger sizes at this level.

Rosenfeld and Gutman (1994) report an effective radius larger than 14 µm needed to trigger C-C and warm rain generation, which is in line with other works (Brenguier and Chaumat, 2001). Furthermore, Pinsky et al. (2001) reports a 3% collision efficiency between collector and collected drops of 60 µm and 10 µm sizes, respectively, while the efficiency increases to 45% for collisions with collected drops of 25 µm sizes. Consequently, the concentration of intermediate sizes (10–30 µm) should be at least one order of magnitude larger than any other size range for an efficient C-C process. Given the dominance of small-sized particles with diameters less than 10 µm and the minimal concentrations of intermediate sizes (10–30 µm) from measurements during both flights, an active C-C process is not achieved. A further inhibiting factor is the limited depth of warm cloud (<1000 m) which is critical for the development of C-C (Johnson, 1993).

For the sub-freezing cloud levels of SF1, the broadening of the particle size distribution (PSD) associated with the decaying turret (-12.4 °C) is primarily attributed to fallout ice (see Fig. 9b) with sizes extending to the millimeter scale. However, for the growing turret with a pure liquid phase at -12.6 °C, the broadening of the size distribution may be a result of the large vertical velocity range (-11.2–17.8 m.s$^{-1}$) or turbulence that increases the supersaturation perturbations (Abade et al., 2018;Grabowski and Abade, 2017;Lasher-trapp et al., 2005). Dust aerosols, acting as ice nuclei, do not appear to be transported to this level, which agrees with the observations of Filioglou et al. (2020) close to the Hajar foothills. For the SF4 sub-freezing levels, broadening is observed for sizes larger than 100 µm extending up to mm-sized irregulars and graupel at -11.2 °C, but at low concentrations within a dominant population of small liquid drops (see Fig. 10a and b). Unlike the case of SF1, dust and pollution aggregates appear to have served as INPs given the strong updrafts (24.4 m.s$^{-1}$) at the -10.6 °C level of SF4.

Several studies show that ultra-giant nuclei (d>40 µm) may serve as precipitation embryos when their concentration is larger than 30 m$^{-3}$ (Bartlett, 1970). The potential sources of such nuclei can be large CCN, sulfate-dominant mineral dust (Wurzler et al., 2000), or simply water insoluble particles serving as coalescence embryos (Lasher-Trapp, 1998). The work of Hoose and

Möhler (2012) suggests that dust particles may form ice particles at T<-15°C. However, they may act as INPs at even higher temperatures depending on their chemical composition, size and concentration (Harrison et al., 2016;Peckhaus et al., 2016).

Reasonable K-feldspar fractions were reported by Kaufmann et al. (2016) from samples collected over Qatar, while no traces were observed in samples from Oman. This may explain the formation of ice irregulars at the sub-freezing levels of SF4 over the southwest near Qatar, given the ice-nucleating properties of K-feldspar.

## 5.3 Implications for hygroscopic cloud-seeding

The general concept of hygroscopic seeding is based on the notion of introducing large artificial (hygroscopic) particles to

compete with smaller naturally occurring aerosols for available cloud LWC. Through this "competition effect", the seeding particles are expected to suppress the activation of smaller background aerosols, rapidly grow into larger drops and trigger C-C (Bruintjes et al., 2012;Cooper et al., 1997). Ghate et al. (2007) studied the impact of introducing giant (salt) seeding aerosols (1–5 µm) into marine stratocumulus clouds using in situ aircraft observations off the central coast of California. Seeding plumes were identified using a threshold of 250 cm$^{-3}$ for the PCASP concentrations compared to a background concentration

of ~80 cm$^{-3}$. They observed a 5-fold increase in the number of large drops (20–40 µm) relative to the background which was attributed to the activation of the seeding GCCN – a small fraction of the total aerosols produced by the flares. Furthermore, Jung et al. (2015) tested even larger seeding particles (1–10 µm) again in marine stratocumulus clouds off the central coast of California and reported a 4-fold increase in the rainfall rate associated with seeding GCCN concentrations of $10^{-2}$–$10^{-4}$ cm$^{-3}$. More recently, Wang et al. (2019) reported on a cloud seeding case study over the eastern coast of Zhejiang, China and

observed the hygroscopic growth of larger-mode seeding particles (>2 µm) up to a limit of ~18 µm drop sizes associated with the competition effect.

The characteristics of the background aerosol population, namely their size, concentration and chemical composition are considered key precursory properties to determine, and potentially improve, the effectiveness of seeding. Segal et al. (2004) report optimum seeding CCN concentrations of 700 cm$^{-3}$ in Mediterranean and extreme continental background conditions.

This concentration is unrealistic in seeding operations and does not account for the impact of large background CCN which is further investigated by their simulations comparing seeded parcels with and without large, natural CCN centered on a diameter of 0.6 µm with concentrations of 0.15 and 0.3 cm$^{-3}$. Their results show a decrease in seeding impact when the large, natural CCN concentrations increased from 0.15 to 0.3 cm$^{-3}$. This was attributed to their competition with the prescribed seeding particles centered on a 10 µm diameter with a concentration of 0.032 cm$^{-3}$. Moreover, the original calculations of Ivanova et

al. (1977) suggest CCN diameters larger than 5 µm to serve as efficient raindrop embryos, while Segal et al. (2007) establish a minimum concentration of 0.025 cm$^3$ for such particles to cause a noticeable increase in warm rain production from a rising cloud parcel under typical conditions in Texas.

The UAE measurements show natural GCCN diameters (5–10 um) concentrations between 0.25–0.15 cm$^{-3}$ which are an order of magnitude larger than the seeding concentration suggested by Segal et al. (2004, 2007). Also, the UAE sub-cloud aerosol
sizes extend from 0.01–100 µm with total concentrations ranging from 500–800 cm$^{-3}$. Hence, all three conceptual models for hygroscopic seeding outlined by Rosenfeld et al. (2010) are applicable to clouds studied over the UAE, namely, accelerating C-C by the competition effect (~1 µm), broadening the cloud drop size distribution by the tail effect (1–10 µm), and introducing ultra-giant seeding particles (>10 µm) to serve as rain drop embryos. These effects need to be thoroughly tested in model simulations based on the observations presented here.

The measurements and analysis provided in this study have important implications for operational seeding activities over the UAE. Our results indicate that relatively large aerosol sizes are already present in the UAE environment - over both the eastern and southwestern region - with comparable sizes to typical hygroscopic flare particles (d<10 µm). Furthermore, the ambient aerosols appear to be hygroscopic in nature with their deliquescence and growth to peak concentrations of ~7 µm sizes at cloud base. This is more pronounced over the southwestern region where mineral aggregates are formed from sea salt and sulfate
particles emitted from local oil refineries (Semeniuk et al., 2015). While the C-C process remained inactive in all observed cases, it is hypothesized that the presence of large dust and pollution aggregates causes a "natural competition effect" as reported by Tessendorf et al. (2021) based on their aircraft observations from the Queensland Cloud Seeding Research Program (QCSRP). Similarly, the modelling work of Segal et al. (2004) indicates a decrease in seeding effects in the presence of large background CCN due to their efficient collision. Hence, given the comparable sizes between the existing GCCN over the UAE
and typical seeding particles, it is unclear if hygroscopic seeding can be effective in these clouds. Modelling studies are needed to investigate whether the concentration and hygroscopicity of the background GCCN are high enough to cause a natural competition effect. Furthermore, modelling studies can help assess the effectiveness of perhaps larger seeding particle sizes (10–15 µm) in augmenting this potentially active natural competition effect and/or in initiating C-C. The modelling work with different seeding materials is in progress and is summarized in Geresdi et al. (2021).

**6 Conclusions**

According to the most recent review on precipitation enhancement research by the World Meteorological Organization (WMO) Expert Team on Weather Modification, a more reliable assessment of the aerosol-cloud-precipitation interaction is needed, particularly using in situ aircraft observations that can validate model results (Flossmann et al., 2019). Located in a regional dust hotspot impacted by air masses originating from five subcontinents over its coastal (west) and mountainous (east) topography, the UAE is considered a "natural laboratory" to study both mesoscale and microscale (aerosol-cloud-precipitation)
processes within a limited geographical area – representative of the larger understudied Arabian Peninsula environment.

Here, we present aerosol and cloud microphysical measurements from research flights targeting two distinct summertime convective regimes in the UAE – orographic convection over the Hajar mountains bordering Oman and non-orographic

convection over the southwestern Saudi border. Despite their different forcing, the thermodynamic and microphysical properties of summertime convective clouds sampled from both regimes are very similar. Sub-cloud aerosol sizes are shown to extend from 10 nm up to 100 µm sizes, with higher concentrations of larger sizes, associated with anthropogenic pollution over the southwest, acting as ultra-giant CCN. The maximum sizes are approximately double those observed over the Sahara desert (Weinzierl et al., 2009). Despite the existence of ultra-giant CCN, no indications of a natural C-C process are observed.

In general, any convective with a cloud depth greater than 500 m is expected to support effective C-C growth (Johnson, 1993). However, the realistically lower LWC within the main body of a convective cloud is more sensitive to cloud base temperature and cloud depth (Warner, 1970). The sampled cloud base temperatures in the UAE (~9 °C) are just below the 10 °C temperature threshold for the onset of an active C-C process (Johnson, 1993;Pruppacher and Klett, 1980), and warm cloud depths never exceed 1000 m in all cases. Furthermore, no indication of C-C is observed within any of the upper levels listed in Table 2 and displayed in Figures 9, 10 and 11. In the upper levels of SF1 (-12.6 and -12.4 °C), a dominant population of liquid drops (d<50 µm) is observed with very few ice particles showing a habit of sector plates (expected by nucleation at -12 °C). LWCs of ~1.4 $g.m^{-3}$ with strong updrafts (~17.8 $m.s^{-1}$) and MVDs less than 20 µm are observed at these sub-freezing levels. Similar observations are also recorded in the upper levels of SF4 with no signs of ice multiplication. The observations show a relatively low concentration of background aerosols in the 10–15 µm size range which may further explain the inactivation of C-C. The results have key implications for ongoing operational cloud-seeding activities over the UAE, which currently rely on hygroscopic material with diameters less than 10 µm. Modelling studies are needed to further assess the influence of aerosols on clouds and precipitation in these clouds, as well as to study implications for hygroscopic cloud-seeding in the UAE. As in Geresdi et al. (2021), model simulations can be initialized (and validated) using the observations provided in this paper to assess the potential of different seeding strategies for augmenting rainfall over this water-stressed region.

**Data availability**

The aircraft observations are archived at the UAE National Center of Meteorology. Readers can request the dataset by contacting research@ncms.ae.

**Author contribution**

ST, CW and YW conceptualized the study. CW, SW and PL supported in data curation and visualization. YW performed the formal analysis and wrote the paper. All co-authors advised on the interpretation of results and were involved in the manuscript editing and discussion.

## Competing interests

The authors declare that they have no conflict of interest.

## Acknowledgements

This material is based on work supported by the National Center of Meteorology, under the UAE Research Program for Rain

Enhancement Science (UAEREP). The work is also supported by the Research Applications Laboratory (RAL) at the National Center for Atmospheric Research (NCAR), sponsored by the National Science Foundation under Cooperative Agreement No. 1852977. The authors thank Dr. Colin Gurganus for advising on data processing and quality control, as well as Prof. István Geresdi and Dr. Sisi Chen for their helpful comments and suggestions.

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

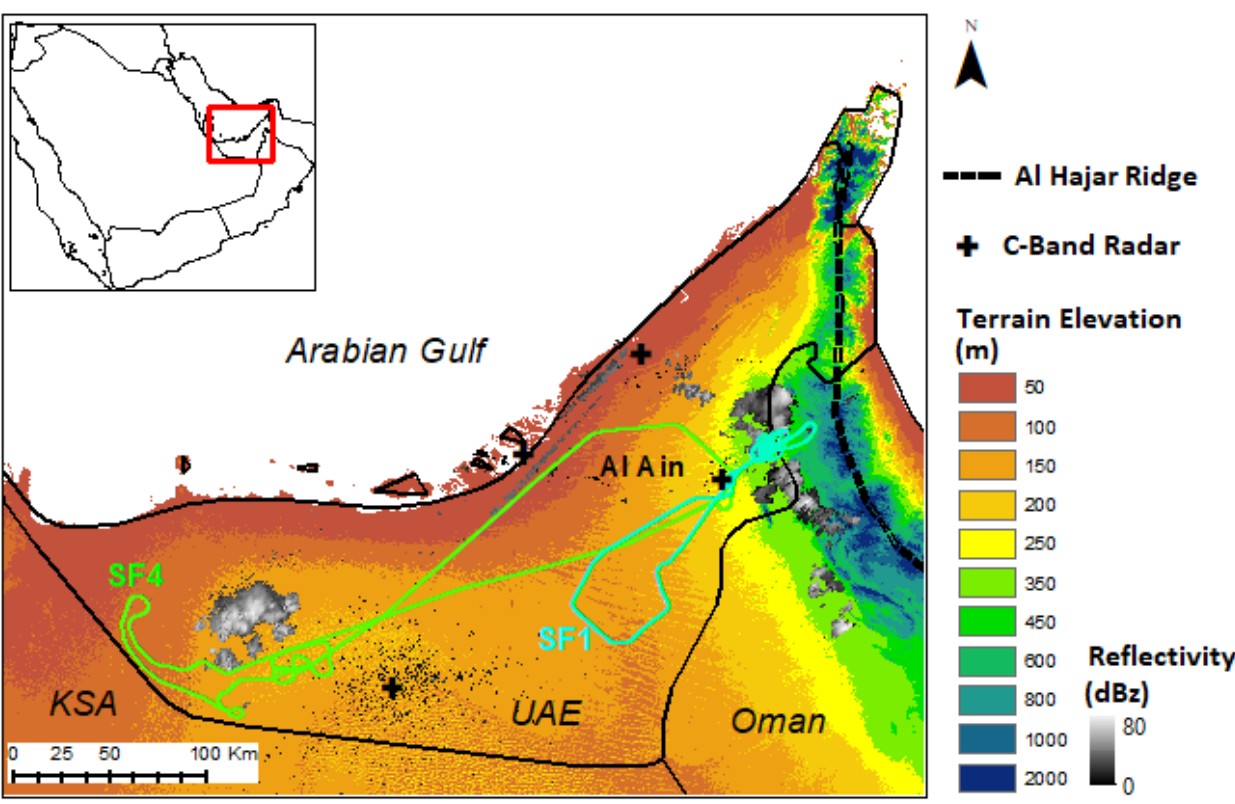

**Figure 1.** Terrain elevation (30 m, ASTER DEM), locations of the C-band radars, reflectivity (dBz) and flight tracks of the August 12th (SF1) and 19th (SF4) cases. Al Ain Airport is in close proximity (~5 km) to the location of the Al Ain radar location shown.


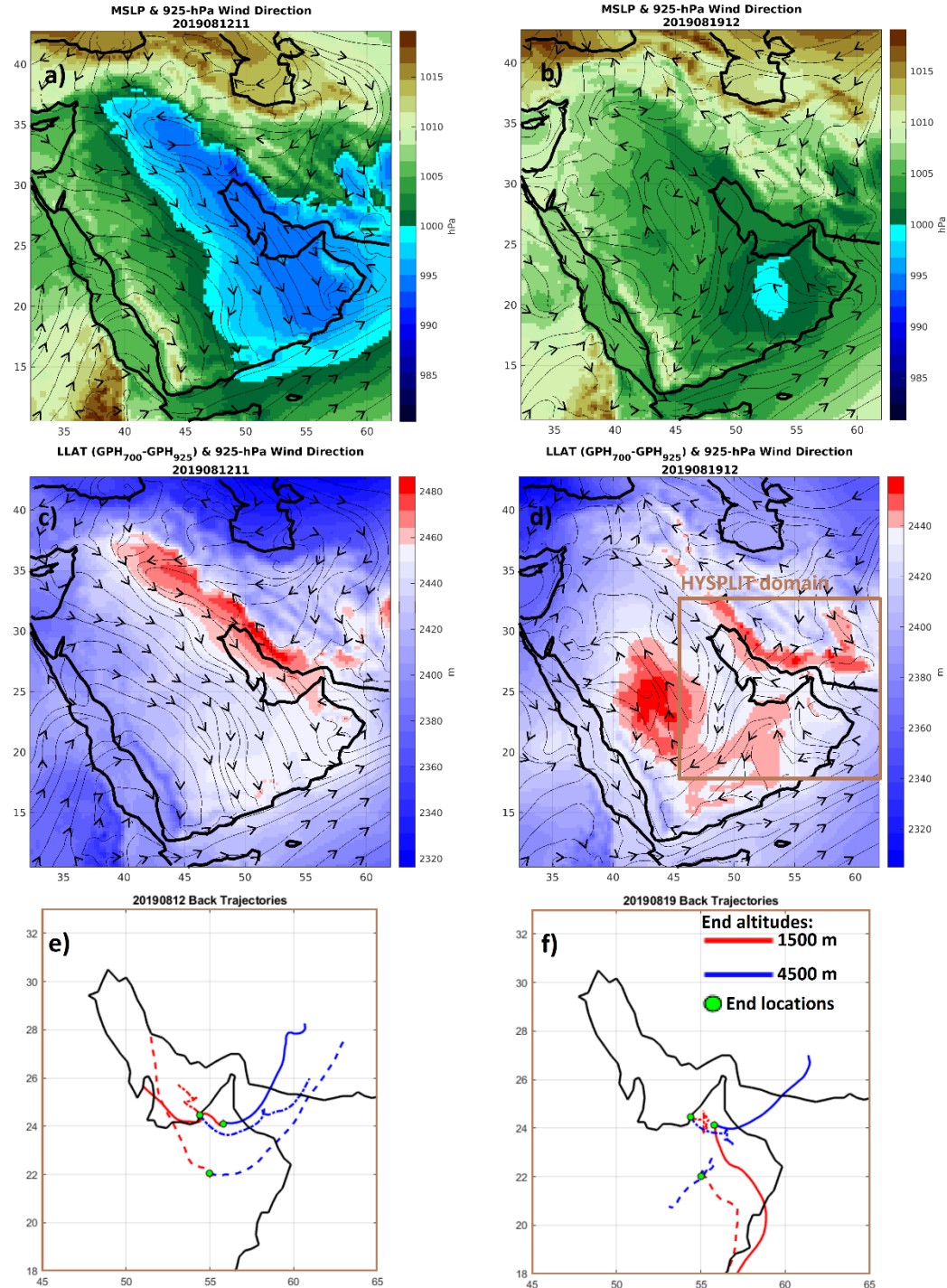

**Figure 2.** MSLP (a,b) and LLAT (c,d) with 925 hPa wind directions from ERA5 reanalysis, and HYSPLIT back-trajectories (e,f) during SF1 and SF4 cases. Red back trajectories had lower altitude end points (1500 m) while blue back trajectories had end points at 4500 m. Different line styles are used to show back-trajectories to each of the three labelled end locations.

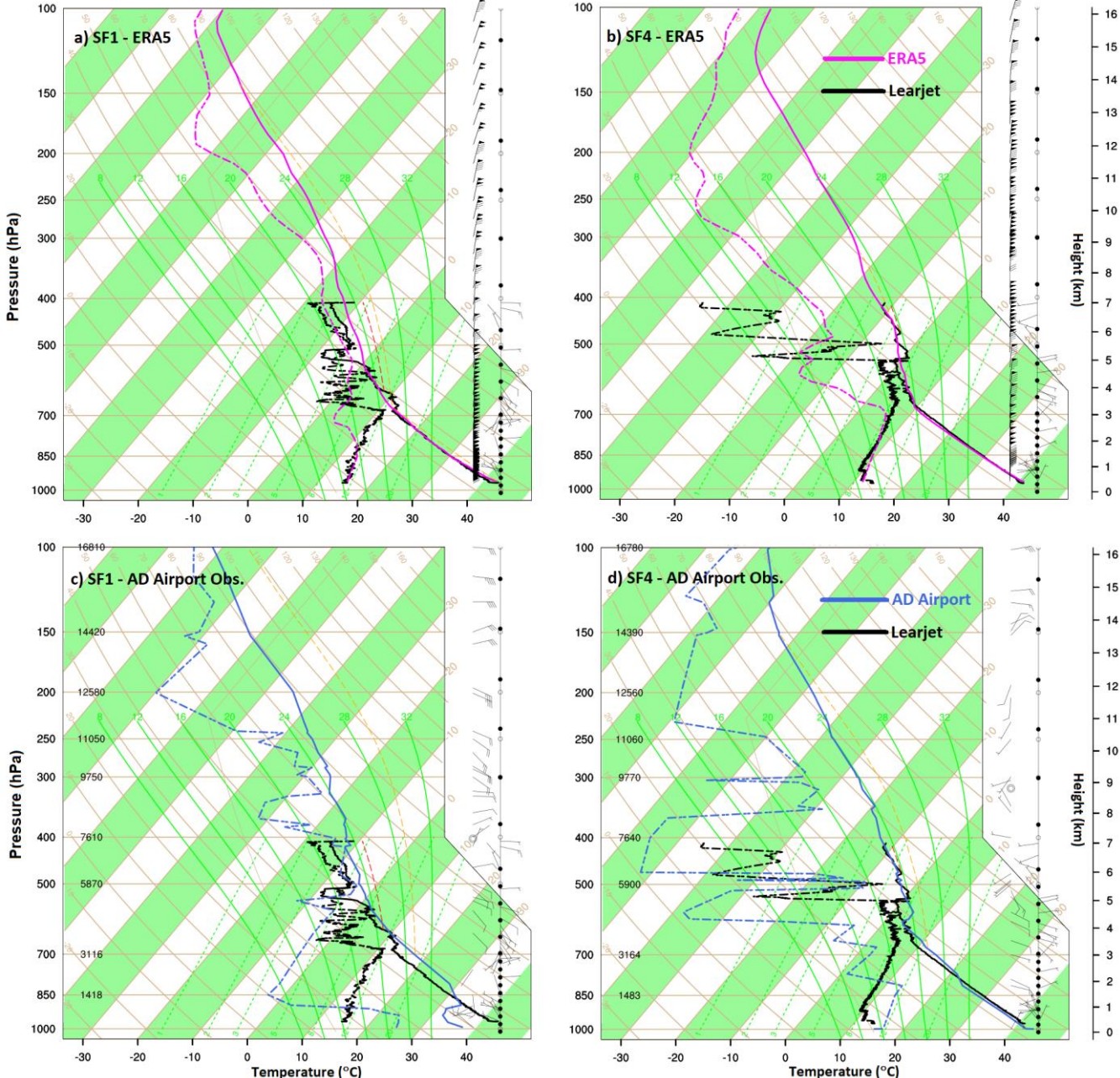

**Figure 3.** Skew-T soundings from SF1 and SF4 descents (limited at 400 hPa level), co-located ERA5 grid points (a,b) and Abu Dhabi Airport (12:00 UTC) observations (c,d).

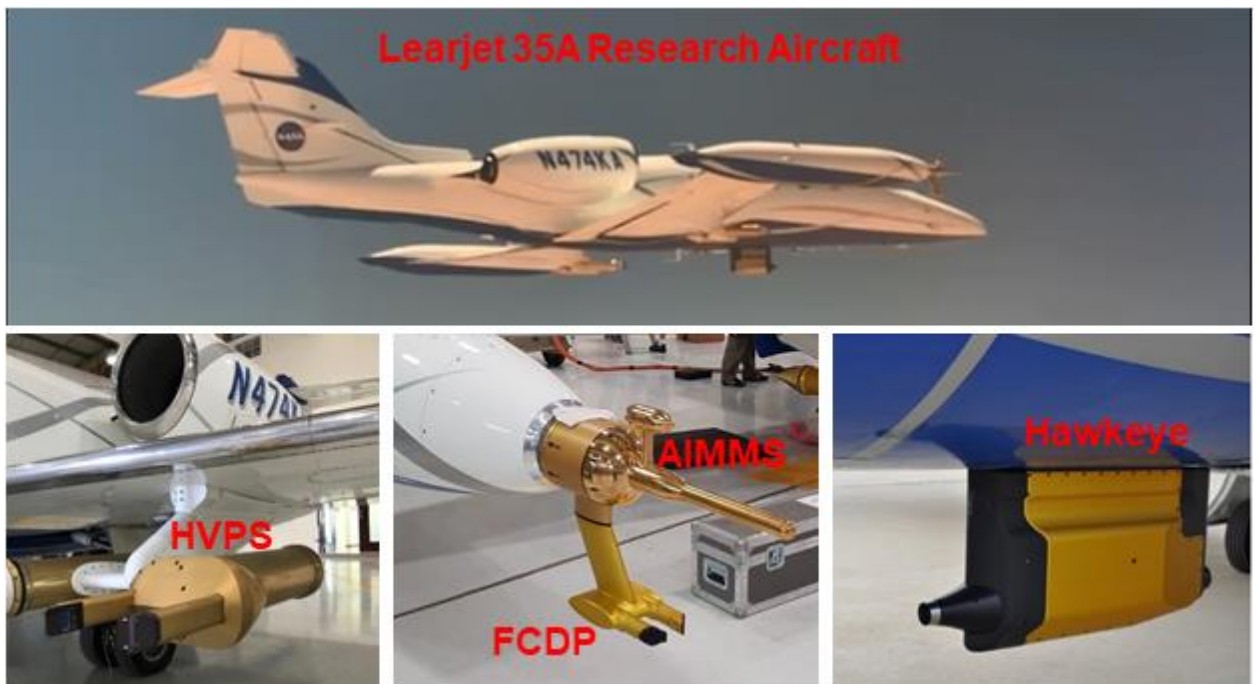


**Figure 4.** Learjet in-flight over Al Ain and selected instruments: HVPS, FCDP, AIMMS and Hawkeye (from left to right).

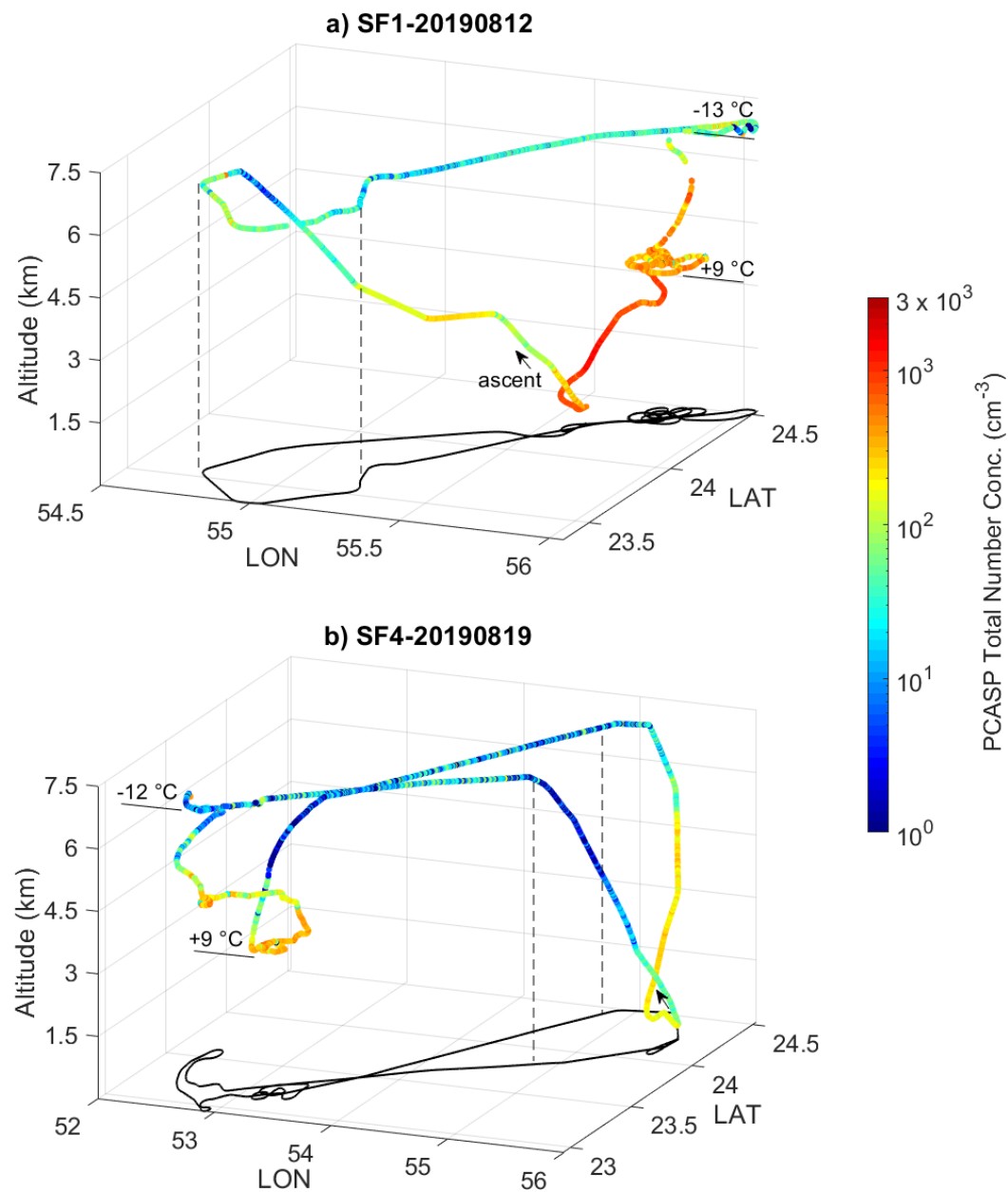

**Figure 5.** 3D flight track of SF1 and SF4 with PCASP total number concentrations (cm$^{-3}$). SF1 penetrations over Al Ain from cloud tops at -13 °C down to 9 °C bases (a), and SF4 penetrations over the southwest from cloud tops at -12 °C down to 9 °C bases (b) are shown.

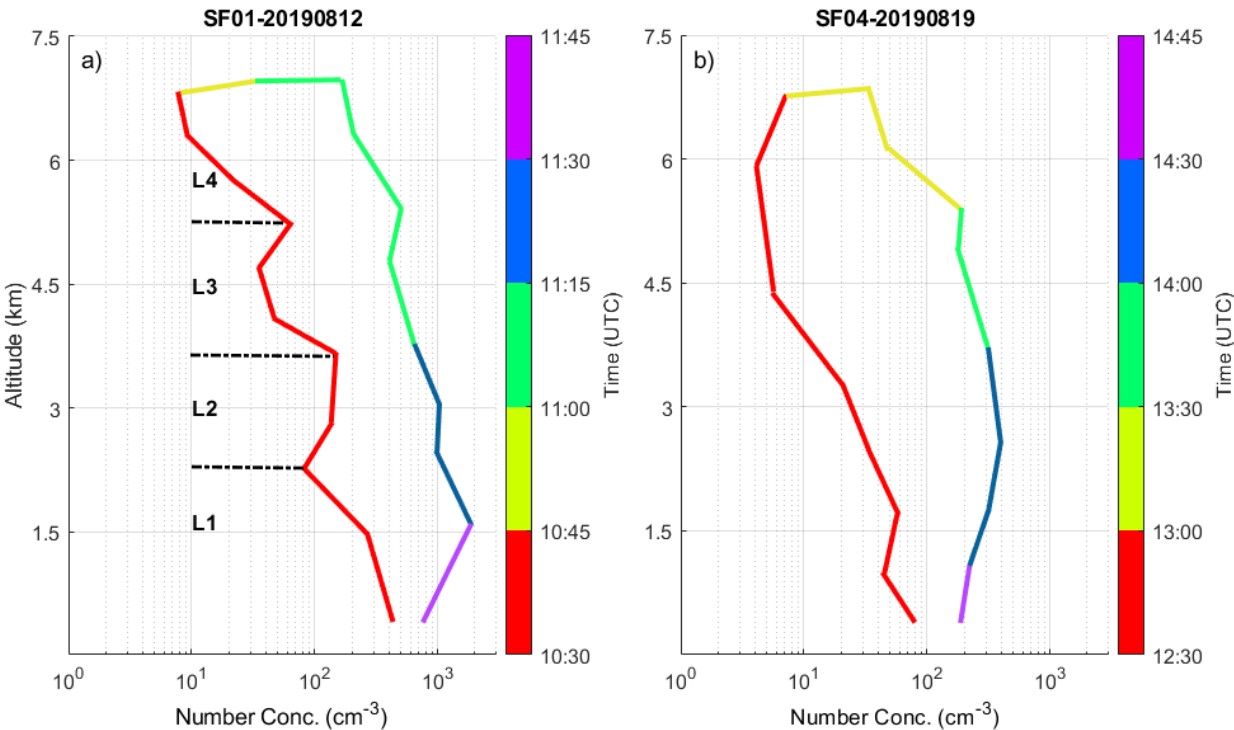

**Figure 6.** Altitude profiles of out-of-cloud PCASP total number concentrations for SF1 (a) and SF4 (b). L1-4 designate
multiple dust layers at levels of significant change in concentration gradients. The profiles are computed as the best-fit from
shape-preserving spline interpolation (Kvasov, 2000) of the full 1-Hz dataset.

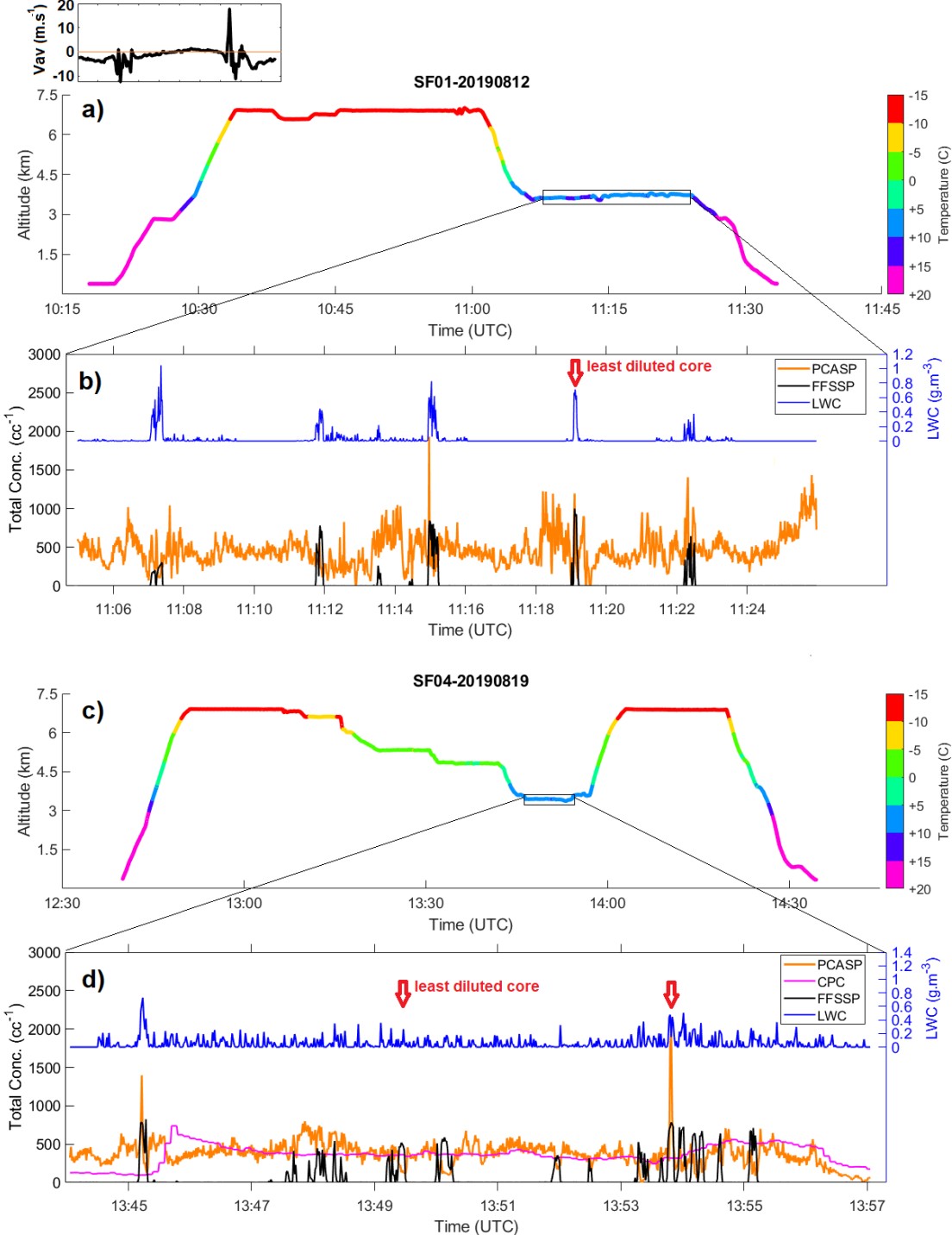

**Figure 7.** Flight profiles for SF1 (a) and SF4 (c) and time series of PCASP, FFSSP, CPC and hot-wire LWC measurements for sub-cloud intervals during SF1 over Al Ain (b) and SF4 over the southwest (d). The inset plot in the top left corner of (a) illustrates the variability in vertical velocity as reported in Table 2 for different levels.

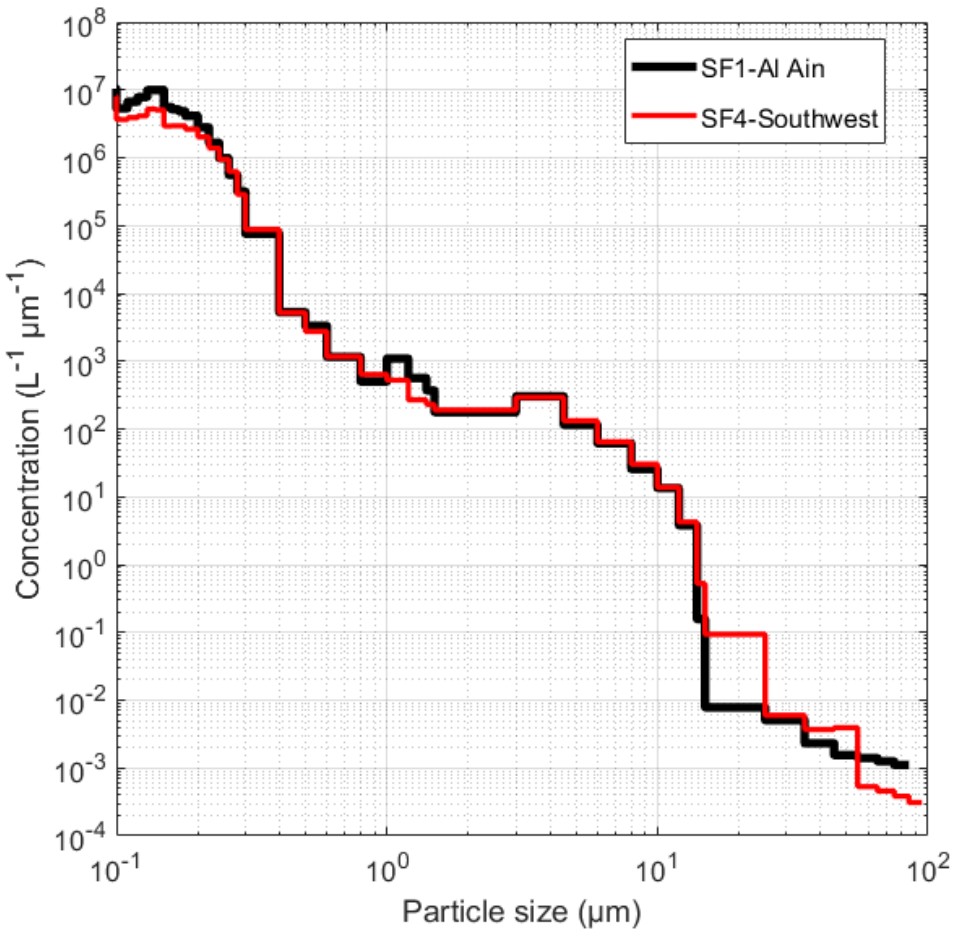

**Figure 8.** Mean sub-cloud aerosol size distributions for during SF1 (Al Ain) and SF4 (southwest).


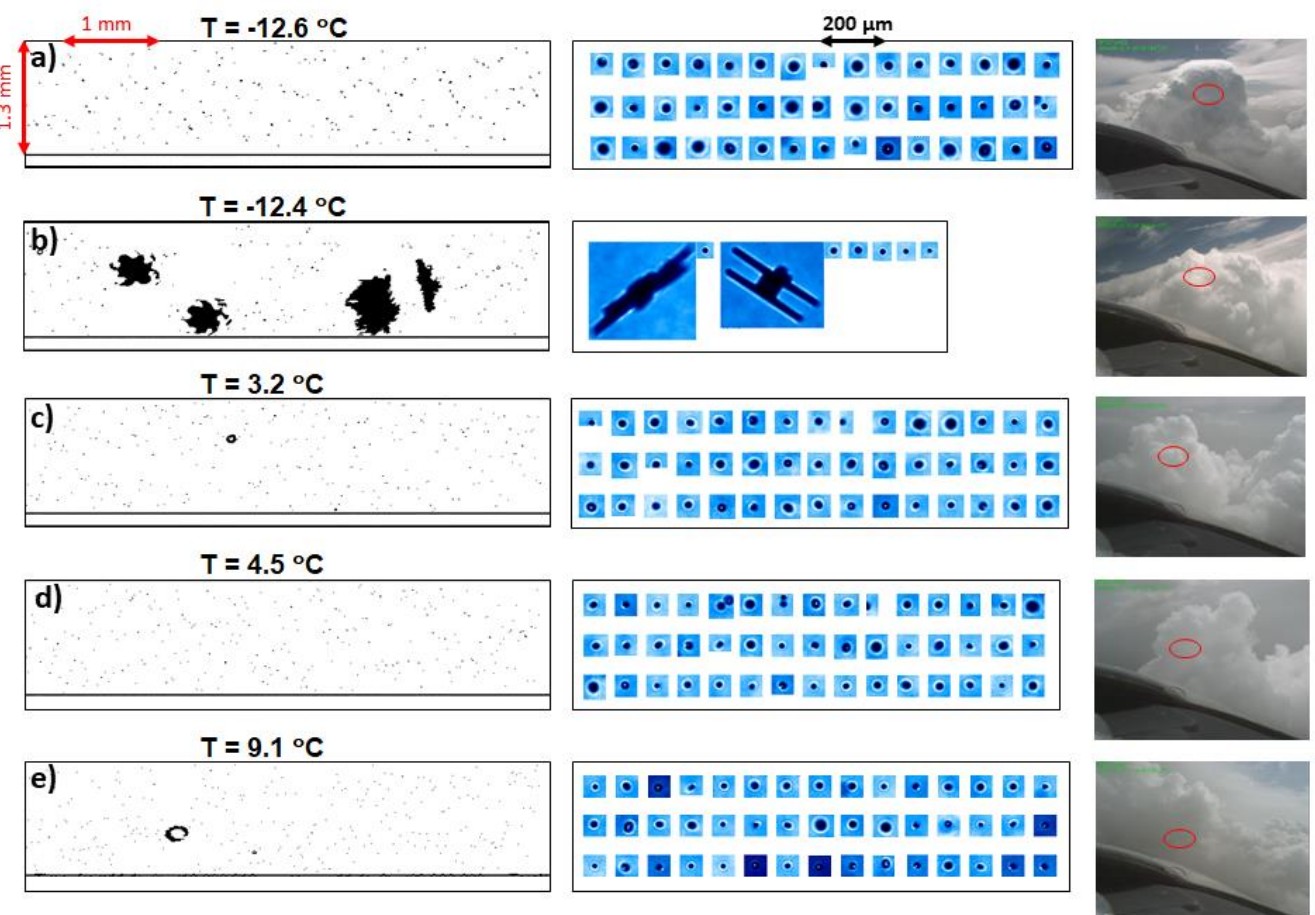

**Figure 9.** SF1 coincident imagery from the 2D-S10, Hawkeye-CPI and cockpit camera. Measurements from each penetration level are listed in Table 2. The cloud penetration locations (red ovals) were determined by visually inspecting video footage from the forward-facing cockpit camera within 1 minute of each cloud approach.


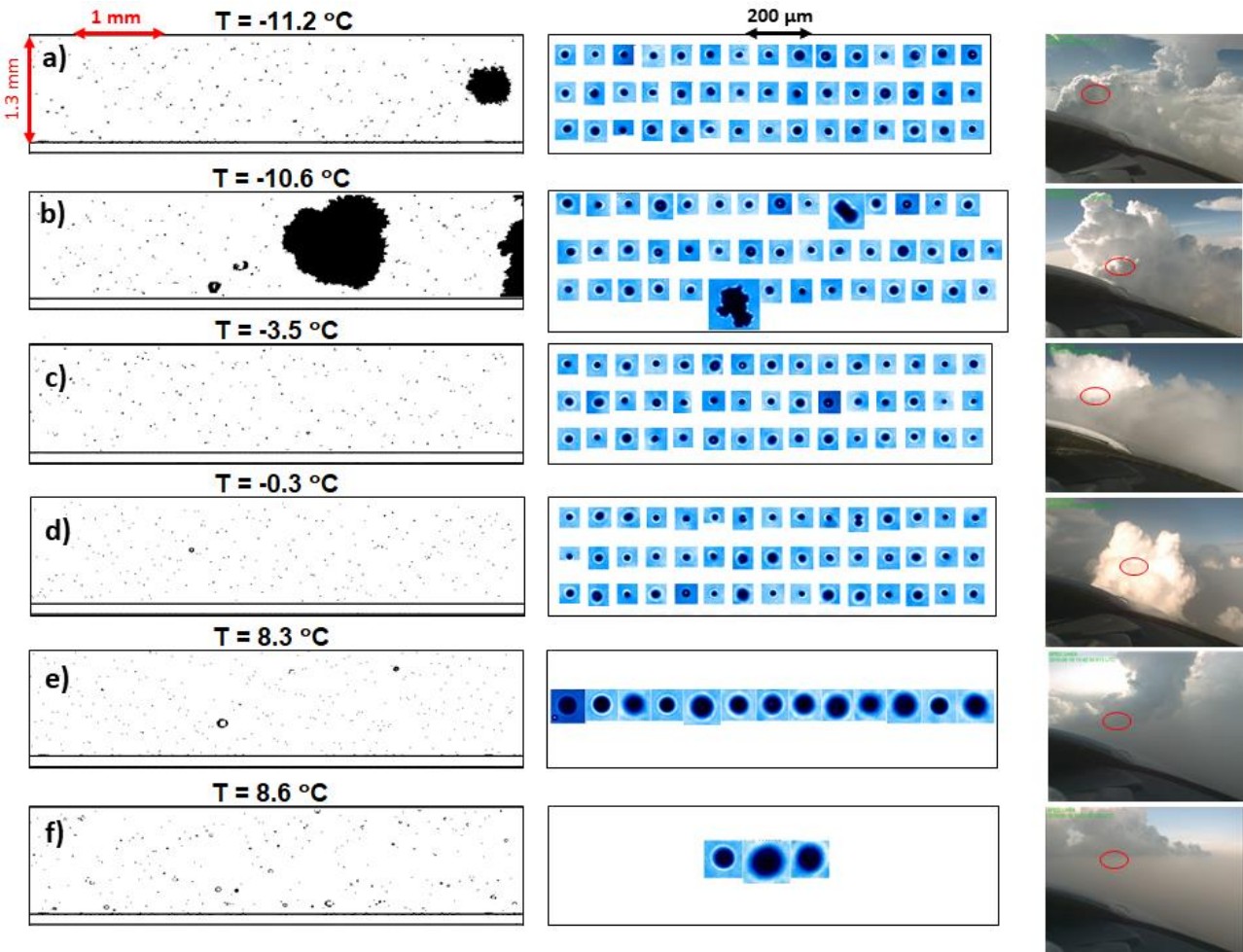

**Figure 10.** Same as Figure 9 but for SF4. Measurements from each penetration level are listed in Table 2.

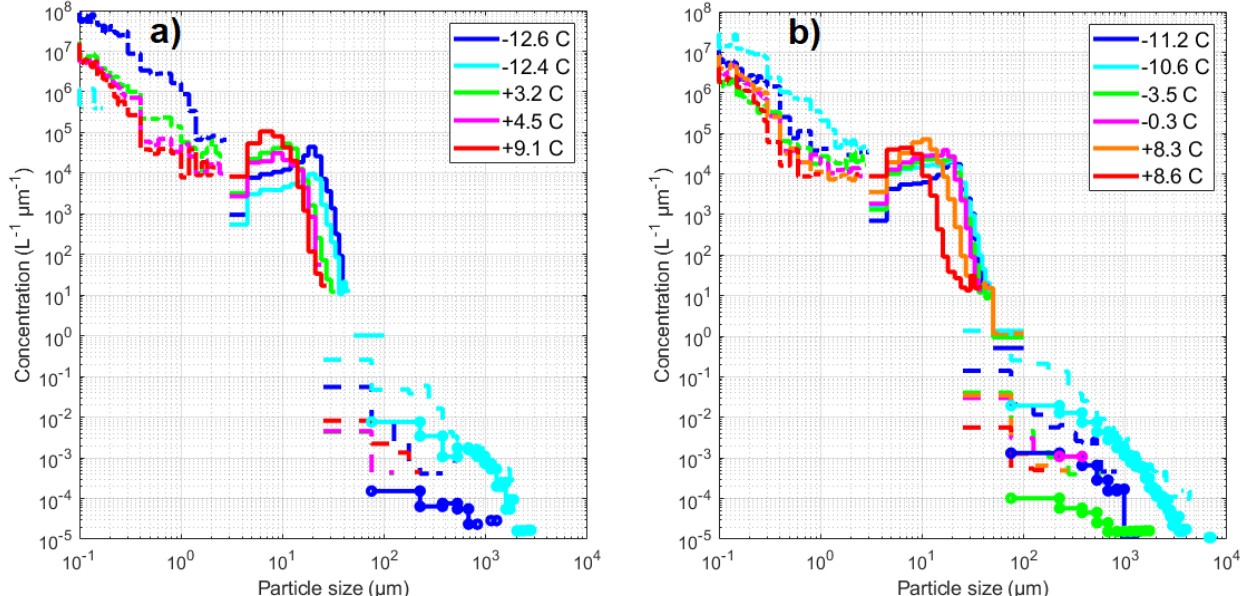


**Figure 11.** Size distributions at multiple temperature levels of growing turrets from the PCASP (dashed), FFSSP (solid), 2D-S50 (dashed) and HVPS (solid with circle markers) measurements during SF1 (a) and SF4 (b).


**Table 1.** List of instruments used for the collection of measurements used herein.

| Instrument | Purpose | Size range | Used here |
|---|---|---|---|
| CPC | Ultra-fine mode aerosol number conc. | 0.01 – 3 µm | ✓ |
| PCASP | Aerosol spectra and conc. | 0.1 – 3 µm | ✓ |
| FCDP | Coarse mode aerosol spectra and conc. | 1.5 – 50 µm | ✓ |
| FCDP-Hawkeye | " | " | |
| FFSSP | Cloud particle spectra and conc. | 3 – 100 µm | ✓ |
| 2D-S10HV[*] | Cloud particle shapes and spectra | 10 – 3000 µm | ✓ |
| 2D-S10HV-NCAR | " | " | |
| 2D-S10V-Hawkeye | " | " | |
| 2D-S50H[**]-Hawkeye | " | 50 – 6400 µm | ✓ |
| CPI-Hawkeye | Cloud particle shapes (ice-phase habit) | 2.3 – 2,300 µm | ✓ |
| HVPS | Precipitation particle shapes and spectra | 150 µm – 2 cm | ✓ |
| Nevzorov Hot-wire | Liquid and total water content | | ✓ |
| AIMMS | Basic meteorological variables | | ✓ |

[*] 10HV indicates 10 µm resolution in both the horizontal (H) and vertical (V) channels.

[**] 50H indicates 50 µm resolution in the horizontal (H) channel.

**Table 2.** Penetration altitude, temperature (T), vertical velocity range (Vv), FFSSP total number concentration (N-FSSP) and MVD values from SF1 and SF4 (see Fig. 9 and 10).

| Altitude (m) | T (°C) | Vv (m.s$^{-1}$) | LWC (g.m$^{-3}$) | N-FFSSP (cm$^{-3}$) | MVD (μm) |
|---|---|---|---|---|---|
| | | SF1 penetration levels (Fig. 9a-e) | | | |
| 7040 | -12.6 | [-11.2 ; 17.8] | 1.4 ± 0.1 | 442 ± 23 | 17.4 ± 0.1 |
| 7050 | -12.4 | [-12.4 ; 1.1] | 1.1 ± 0.1 | 223 ± 18 | 19.2 ± 0.1 |
| 4660 | 3.2 | [-1.3 ; 6.2] | 1.2 ± 0.1 | 619 ± 28 | 10.9 ± 0.2 |
| 4430 | 4.5 | [-2.9 ; 6.9] | 1.2 ± 0.1 | 621 ± 16 | 9.6 ± 0.6 |
| 3720 | 9.1 | [-2.4 ; 3.1] | 0.8 ± 0.05 | 800 ± 32 | 8.7 ± 0.3 |
| | | SF4 penetration levels (Fig. 10a-f) | | | |
| 6710 | -11.2 | [-2.6 ; 4] | 1.2 ± 0.1 | 321 ± 28 | 19.4 ± 0.2 |
| 6700 | -10.6 | [-7.6 ; 24.4] | 1.3 ± 0.1 | 778 ± 42 | 19.5 ± 0.4 |
| 5400 | -3.5 | [-6.1 ; 4.6] | 1.1 ± 0.1 | 479 ± 21 | 17.1 ± 0.3 |
| 4910 | -0.3 | [-3.4 ; 11.4] | 1.3 ± 0.1 | 470 ± 33 | 13.2 ± 0.2 |
| 3620 | 8.3 | [-2.3 ; 3.3] | 0.5 ± 0.04 | 753 ± 29 | 10.6 ± 0.4 |
| 3500 | 8.6 | [-0.5 ; 3.2] | 0.2 ± 0.03 | 541 ± 26 | 7.1 ± 0.1 |