# Peer review of "Analysis of aerosol-cloud interactions and their implications for precipitation formation using aircraft observations over the United Arab Emirates"

_Atmospheric Chemistry and Physics, 2021_

## Author Comment (AC1)

**Response to Reviewer 1 Comments**

This study analyzes in-situ aircraft measurements in summer convective clouds over the UAE. Two cases are analyzed – one is orographically triggered, and the other is over a flat desert terrain. The measurements document the sub-cloud aerosols and the vertical evolution of cloud drop size distribution and hydrometeors in the convective elements between cloud base near height of 3.5 km up to 7 km. The data are analyzed in the context of assessment of cloud seeding potential for rain enhancement.

The aircraft data analysis and its interpretation have numerous issues which have to be resolved before this paper can be accepted for publication.

We thank Reviewer 1 for the many insightful comments and suggestions. Below is our point-by-point response to the provided comments.

There are several questions about data quality and inconsistencies:

1) Comment 1: 7d shows generally a larger concentration of PCASP than CPC particles, which is physically impossible. The CPC concentrations are always more numerous by a wide margin, because the PCASP particles are a subset of the CPC particles. Therefore, the claim made in lines 230-232 that CPC concentrations greater than PCASP "allude to the hygroscopic nature of the ultra-fine background aerosols" is unfounded.

- Response to Comment 1: We thank the reviewer for raising this point. The CPC and PCASP instruments have different sampling mechanisms – the CPC provides counts of ultra-fine mode aerosols (0.01–3 µm) through deliberate condensation of intercepted particles to reach sizes detectable by a laser counter, while the PCASP measures dry aerosol particles sizes (0.1–3 µm) via particle-light scattering. The CPC generally records less variability as it samples aerosols at a significantly larger volumetric rate (50 $cm^3$/s) compared to that of the PCASP (1 $cm^3$/s) (Cai et al., 2013;Wiedensohler et al., 2012).

  The time series in Figure 7d shows larger concentrations of PCASP compared to CPC particles during the initial interval (13:44–13:45:30), and vice versa for the final interval (13:54–13:57). However, the CPC particle concentrations fall within the 20% uncertainty margin (Rosenberg et al., 2012) of the PCASP particle concentrations during the inner interval (13:47–13:54).

  Given the different sampling mechanisms and inconsistencies between the observations from the two instruments, we agree with the reviewer that the hygroscopic nature of the ultra-fine background aerosols remains questionable. Nevertheless, the comparable measurements during the majority of the flight time (13:47–13:54) suggests a smaller concentration of ultra-fine (0.01–0.1 µm) compared to larger particles (0.1–3 µm).

- Changes to Manuscript: Removed Lines 229–232 and added Lines 239–245: "The CPC generally records less variability as it samples aerosols at a significantly larger volumetric rate (50 $cm^3$/s) compared to that of the PCASP (1 $cm^3$/s) (Cai et al., 2013;Wiedensohler et al., 2012). Figure 7d shows larger concentrations of PCASP compared to CPC particles during the initial interval (13:44–13:45:30), and vice versa for the final interval (13:54–13:57). However, the CPC particle concentrations fall within the 20% uncertainty margin (Rosenberg et al., 2012) of the PCASP particle concentrations during the inner interval (13:47–13:54). The comparable measurements during the majority of the SF4 flight

track over the southwest (13:47–13:54) suggests a smaller concentration of ultra-fine (0.01–0.1 µm) compared to larger particles (0.1–3 µm)."

2) Comment 2: The maximum FFSSP concentrations are even larger than the sub-clouds PCASP and CPC concentrations, which is again physically impossible.

- Response to Comment 2: The FFSSP concentrations are always less than PCASP concentrations during the SF1 sub-cloud time series (see Figure 7b). However, in the case of the dustier sub-cloud conditions of SF4 (Figure 7d), there are instances where the peak FFSSP concentrations are larger than the PCASP (and CPC) concentrations, with relative differences less than 20%. Differences in flow rate, refractive index, and relative humidity-dependent errors introduce inconsistencies in the calibration curves of the optical sizing instruments with an average uncertainty of 28% considered acceptable for the inter-comparison of their measurements (Moore et al., 2004;Reid et al., 2003).

- Changes to Manuscript: The above paragraph is added in Lines 233–238.

3) Comment 3: Furthermore, the FCDP concentrations are even larger than the FFSSP concentrations by up to a factor of 2.5, as evident in line 246. Nevertheless, the authors attempt to provide a physical explanation arguing that most of the cloud droplets are smaller than a diameter of 6 µm (lines 247-249), without showing it.

Response to Comment 3: Figure R1 below shows the mean sub-cloud particle size distributions (limited to sizes < 20 µm) from the PCASP, FFSSP and FCDP instruments from (a) SF1 and (b) SF4. In both cases, the FCDP overlaps well with the tail of the PCASP measurements (1.5–3 µm) and captures sizes smaller than 6 µm which are missed by the FFSSP. For sizes of 6 µm and beyond, the FFSSP and FCDP spectra appear to converge.

The FFSSP has legacy optics demonstrated in previous campaigns (Lawson et al., 2015;Lawson et al., 2017) and while the FCDP is billed to have improved optics that are more sensitive to the smallest particle sizes, the processing depends on cloud drop concentration and subjective selection of processing variables. As outlined in Table 1, the FCDP is used for coarse mode aerosols, while the legacy FFSSP is used for in-cloud measurements (Section 5.2). However, we state the FCDP in-cloud measurements alongside the FFSSP measurements to indicate the variability between the instruments which requires further work beyond the scope of the current paper. The comparison of measurements from the two instruments remains relevant for calibration purposes, particularly for their operation in highly polluted environments such as that in the UAE.

[Figure]

Figure R1. Mean sub-cloud particle size distributions (sizes < 20 μm) from the PCASP, FFSSP and FCDP instruments from (a) SF1 and (b) SF4.

The physical interpretation of the measurements has the following issues:

4) Comment 4: Lines 275-276; 289-291: The broadening of the drop size distributions is interpreted as an evidence for the hygroscopic characteristics of background ultrafine aerosols. It is also ascribed to the effects of strong updraft and turbulence. However, examination of Fig. 11 shows that the peak of the FFSSP concentrations increases systematically with height and with decreasing temperature. This has to be this way in a hypothetical adiabatic cloud without coalescence, regardless of all the considerations raised by the authors. The increase beyond an adiabatic rate would be potentially explained by coalescence and the effects of turbulence and drop spectrum shape on it.

Response to Comment 4: Avoiding entrainment effects is particularly challenging in the dry environment of the UAE, where even the most undiluted penetrations are contaminated by downdrafts (see Table 2 in the manuscript). Figure R2 below shows the observed (FFSSP, FCDP and hotwire) and calculated adiabatic liquid water contents at different temperatures from (a) SF1 and (b) SF4. The penetrated clouds are generally sub-adiabatic near their bases (~9 °C).

[Figure]

Figure R2. FFSSP, FCDP, hotwire and the calculated adiabatic liquid water contents at different temperatures from (a) SF1 and (b) SF4.

Hence, the updraft is strongly influenced by entrainment as shown by the sub-adiabatic LWC in both cases. Any broadening is expected from condensational growth. To further investigate this, Figure R3 below shows a plot of temperature versus effective radius (Reff) from averages of cloud penetrations during the UAE flight campaign and two previous campaigns over the southeastern United States (SEUS) and Philippine Sea which co-authors Lawson and Woods participated in. The gray shading is the region where Freud and Rosenfeld (2012) predict active coalescence would occur. There is a consensus in the literature that a value of Reff > ~ 12 to14 µm is required to initiate the formation of drops that trigger the coalescence process (Rosenfeld and Gutman, 1994;Gerber, 1996;Andreae et al., 2004;Freud and Rosenfeld, 2012). Thus, the larger the value of Reff near cloud base favors development of coalescence higher in the cloud. Reff values from the UAE measurements never extend beyond the threshold value for an active collision-coalescence, unlike those recorded over the SEUS and Philippine Sea.

[Figure]

Figure R3. Temperature versus effective radius (Reff) from averages of cloud penetrations from the UAE campaign and two previous campaigns the co-authors led the southeastern United States (SEUS) and Philippine Sea. The gray shading is the region where Freud and Rosenfeld (2012) predict active coalescence would occur.

5) Comment 5: Lines 312-314: The mere existence of aerosols at the size range of hygroscopic flare particles does not serve as indication for seeding potential. The important property is the concentrations of these background aerosols compared to the concentrations of flare aerosols which is required for achieving a desirable extent of rain enhancement, at least in theory according to previous research. The authors state that more simulations should be conducted, but this does exempt them from the need to go deeper in assessing the already available knowledge about that. Some of that knowledge is given in Segal et al. (2004), which the authors referenced erroneously in another context.

• Response to Comment 5: We agree with the reviewer on the need to elaborate further on the role of concentration (and size) of natural CCN concentrations in relation to seeding potential.

The characteristics of the background aerosol population, namely their size, concentration and chemical composition are considered key precursory properties to determine, and potentially improve, the effectiveness of seeding. Segal et al. (2004) report optimum seeding CCN concentrations of 700 cm$^{-3}$ in Mediterranean and extreme continental background conditions. This concentration is unrealistic in seeding operations and does not account for the impact of large background CCN which is further investigated by their simulations comparing seeded parcels with/without large, natural CCN centered on a diameter of 0.6 μm with concentrations of 0.15 and 0.3 cm$^{-3}$. Their results show a decrease in seeding impact when the large, natural CCN concentrations increased from 0.15 to 0.3

cm$^{-3}$. This was attributed to the competition with the prescribed seeding particles centered on a 10 µm diameter with a concentration of 0.032 cm$^{-3}$. Moreover, the original calculations of Ivanova et al. (1977) suggest CCN diameters larger than 5 µm serve as efficient raindrop embryos, while Segal et al. (2007) establish a minimum concentration of 0.025 cm$^3$ for such particles to cause a noticeable increase in warm rain production from a rising cloud parcel under typical conditions in Texas.

The UAE measurements show natural GCCN diameters (5–10 um) concentrations between 0.25–0.15 cm$^{-3}$ which are an order of magnitude larger than the seeding concentration suggested by Segal et al. (2004, 2007). Also, the UAE sub-cloud aerosol sizes extend from 0.01–100 µm with total concentrations ranging from 500–800 cm$^{-3}$. Hence, all three conceptual models for hygroscopic seeding outlined by Rosenfeld et al. (2010) are applicable to clouds studied over the UAE, namely, accelerating collision-coalescence by the competition effect (~1 µm), broadening the cloud drop size distribution by the tail effect (1–10 µm), and introducing ultra-giant seeding particles (>10 µm) to serve as rain drop embryos. These effects need to be thoroughly tested in model simulations based on the observations presented here.

The modeling work with different seeding material is in progress and is summarized in Geresdi et al. (2021).

- Changes to Manuscript: Added first two paragraphs in the above response to Section 5.3 Lines 332–348.

6) Comment 6: Lines 314-316: The peak concentrations at cloud base is much smaller than 20 µm. According to Fig. 11 the peak drop concentrations at cloud base reaches only 7 µm in both flights.

- Response to Comment 6: We agree with the reviewer but the original statement was "above" (i.e. sub-cloud) instead of "at" cloud base. We revised the statement for further clarity.

- Changes to Manuscript: revised Lines 353–355: "the ambient aerosols appear to be hygroscopic in nature with their deliquescence and growth to peak concentrations of ~7 µm sizes at cloud base."

7) Comment 7: Lines 318-319: To claim that the large salt and dust particles causes a significant competition effect requires a quantitative assessment of the observed concentrations of these large particles compared to the theoretical concentrations that would have such an effect with a magnitude of practical importance. Again, there is available relevant knowledge that should be referenced and discussed.

- Response to Comment 7:

In line with our response to comment 5, Segal et al. (2004) simulated seeded parcels with and without large, natural CCN centered on a diameter of 0.6 µm with concentrations of 0.15 and 0.3 cm$^{-3}$. Their results show a decrease in seeding impact when the large, natural CCN concentrations increased from 0.15 to 0.3 cm$^{-3}$. This was attributed to their competition with the prescribed seeding particles (centered on a 10 µm diameter with a concentration of 0.032 cm$^{-3}$). Also, the original calculations of Ivanova et al. (1977) suggest CCN diameters larger than 5 µm to serve as efficient raindrop embryos, while Segal et al. (2007) establish a minimum concentration of 0.025 cm$^3$ for such particles to cause a noticeable increase in warm rain production from a rising cloud parcel under typical conditions in

Texas. The UAE measurements show natural GCCN diameters (5–10 um) concentrations between 0.25–0.15 cm$^{-3}$ which are an order of magnitude larger than the seeding concentration suggested by Segal et al. (2004, 2007).

Again, all three conceptual models for hygroscopic seeding outlined by Rosenfeld et al. (2010) are applicable to clouds studied over the UAE, namely, the competition effect (~1 μm), tail effect (1–10 μm), and rain embryo (>10 μm) effect. These effects need to be thoroughly tested in model simulations based on observations from the current paper. The modeling work with different seeding material is in progress and is summarized in Geresdi et al. (2021).

- Changes to Manuscript: addressed in response to comment 5 above.

8) Comment 8: Line 320-321: Segal et al. (2004) did NOT claim that seeding effect is smaller with larger background CCN concentrations. He rather claimed that the seeding effect is larger with a greater amount of seeding material, up to an optimal point.

- Response to Comment 8: Following our response to comment 7, Segal et al. (2004) state (Page 30, Paragraph 3): "*Note that, in the presence of significant concentrations of natural large CCN, the seeding effect decreases significantly due to the efficient collision process initiated by droplets growing on these CCN. One can say that, in these cases, clouds have already been seeded by natural, large CCN (Rosenfeld et al., 2002)*".

9) Comment 9: Lines 21; 341-344: The authors state that there is no collision and coalescence (CC) in the lowest 1000 m of the clouds. However, the important question is the extent of CC at all heights, and is there any evidence of CC at any height? This is very important, because precipitation can often initiate as supercooled rain drops. Furthermore, the abundance of CC promotes ice multiplication, further accelerating the precipitation initiation. This should be assessed to the possible extent from the aircraft data.

- Response to Comment 9: No indication of C-C is observed within any of the upper levels listed in Table 2 and displayed in Figures 9, 10 and 11. In the upper levels of SF1 (-12.6 and -12.4 °C), a dominant population of liquid drops (d<50 μm) is observed with very few ice particles showing a habit of sector plates (as expected by nucleation at -12 °C). LWCs of ~1.4 g.m$^{-3}$ with strong updrafts (~17.8 m.s$^{-1}$) and MVDs less than 20 μm are observed at these sub-freezing levels. Similar observations are also recorded in the upper levels of SF4 with no signs of ice multiplication.

- Changes to Manuscript: The above paragraph is added in Lines 384–388.

10) Comment 10: Line 75: Lawson et al. (2019) is missing in the reference list.

- Response to Comment 10: This reference was listed in Line 466 in the original manuscript – now Line 523 in the revised manuscript.

**Authors' References**

- Andreae, M. O., Rosenfeld, D., Artaxo, P., Costa, A. A., Frank, G. P., Longo, K. M., and Silva-Dias, M. A. F.: Smoking rain clouds over the Amazon, Science, 303, 1337-1342, 2004.
- Cai, Y., Snider, J. R., and Wechsler, P.: Calibration of the passive cavity aerosol spectrometer probe for airborne determination of the size distribution, Atmospheric Measurement Techniques, 6, 2349-2358, 2013.
- Freud, E., and Rosenfeld, D.: Linear relation between convective cloud drop number concentration and depth for rain initiation, Journal of Geophysical Research: Atmospheres, 117, 2012.
- Gerber, H.: Microphysics of marine stratocumulus clouds with two drizzle modes, Journal of Atmospheric Sciences, 53, 1649-1662, 1996.
- Geresdi, I., Chen, S., Wehbe, Y., Bruintjes, R., Lee, J., Tessendorf, S., Weeks, C., Sarkadi, N., Rasmussen, R. M., and Grabowski, W.: Sensitivity of the Efficiency of Hygroscopic Seeding on the Size Distribution and Chemical Composition of the Seeding Material, 101st American Meteorological Society Annual Meeting, 2021,
- Ivanova, E., Kogan, Y., Mazin, I., and Permyakov, M.: The ways of parameterization of condensation drop growth in numerical models, Izv. Atmos. Oceanic Phys, 13, 1193-1201, 1977.
- Lawson, P., Gurganus, C., Woods, S., and Bruintjes, R.: Aircraft observations of cumulus microphysics ranging from the tropics to midlatitudes: Implications for a "new" secondary ice process, Journal of Atmospheric Sciences, 74, 2899-2920, 2017.
- Lawson, R. P., Woods, S., and Morrison, H.: The microphysics of ice and precipitation development in tropical cumulus clouds, Journal of the Atmospheric Sciences, 72, 2429-2445, 2015.
- Moore, K., Clarke, A., Kapustin, V., McNaughton, C., Anderson, B., Winstead, E., Weber, R., Ma, Y., Lee, Y., and Talbot, R.: A comparison of similar aerosol measurements made on the NASA P3-B, DC-8, and NSF C-130 aircraft during TRACE-P and ACE-Asia, Journal of Geophysical Research: Atmospheres, 109, 2004.
- Reid, J. S., Jonsson, H. H., Maring, H. B., Smirnov, A., Savoie, D. L., Cliff, S. S., Reid, E. A., Livingston, J. M., Meier, M. M., and Dubovik, O.: Comparison of size and morphological measurements of coarse mode dust particles from Africa, Journal of Geophysical Research: Atmospheres, 108, 2003.
- Rosenberg, P., Dean, A., Williams, P., Dorsey, J., Minikin, A., Pickering, M., and Petzold, A.: Particle sizing calibration with refractive index correction for light scattering optical particle counters and impacts upon PCASP and CDP data collected during the Fennec campaign, Atmospheric Measurement Techniques, 5, 1147-1163, 2012.
- Rosenfeld, D., and Gutman, G.: Retrieving microphysical properties near the tops of potential rain clouds by multispectral analysis of AVHRR data, Atmospheric research, 34, 259-283, 1994.
- Rosenfeld, D., Lahav, R., Khain, A., and Pinsky, M.: The role of sea spray in cleansing air pollution over ocean via cloud processes, Science, 297, 1667-1670, 2002.
- Rosenfeld, D., Axisa, D., Woodley, W. L., and Lahav, R.: A quest for effective hygroscopic cloud seeding, Journal of applied meteorology and climatology, 49, 1548-1562, 2010.
- Segal, Y., Khain, A., Pinsky, M., and Rosenfeld, D.: Effects of hygroscopic seeding on raindrop formation as seen from simulations using a 2000-bin spectral cloud parcel model, Atmospheric Research, 71, 3-34, 2004.
- Segal, Y., Pinsky, M., and Khain, A.: The role of competition effect in the raindrop formation, Atmospheric research, 83, 106-118, 2007.
- Wiedensohler, A., Birmili, W., Nowak, A., Sonntag, A., Weinhold, K., Merkel, M., Wehner, B., Tuch, T., Pfeifer, S., and Fiebig, M.: Mobility particle size spectrometers: harmonization of technical

standards and data structure to facilitate high quality long-term observations of atmospheric particle number size distributions, Atmospheric Measurement Techniques, 5, 657-685, 2012.

---

## Author Comment (AC2)

**Response to Reviewer 2 Comments**

**General comments**

This manuscript presents a study of aerosol (especially giant particles such as mineral dust) & cloud interactions over UAE regions by using aircraft measurements. Comparative analysis from penetrated sampling by two research flights gives important information on cloud microphysical characterizes and precipitation formation mechanisms over this region. Among many recent studies on aerosol-cloud interaction, this research provides a new insight into understanding the influence of aerosols on cloud and precipitation processes, as well as its application for hygroscopic cloud seeding. This paper is overall well-written, but some scientific discussions tend to draw conclusions quickly without a strong statement, especially the linkage between aerosol properties and cloud microphysical process in section 5. In general, minor revisions are needed before the acceptance of this manuscript. Below listed are the comments and suggestions.

We thank Reviewer 2 for the many insightful comments and suggestions. Below is our point-by-point response to the provided comments.

**Specific comments**

1)  Comment 1: In section 1: This part is a review of the roles that aerosols play in the cloud microphysical process. However, it lacks some important introductions such as the aerosol effect on precipitation or its application on hygroscopic seeding, as the title includes "… aerosol-cloud interactions … precipitation formation …". Please give a literature review about research that has been conducted in association with aerosols (especially giant CCN) as an agent of cloud seeding (you can put this part in this section or section 5.3):

    - *Jung, E., Albrecht, B. A., Jonsson, H. H., Chen, Y.-C., Seinfeld, J. H., Sorooshian, A., Metcalf, A. R., Song, S., Fang, M., and Russell, L. M.: Precipitation effects of giant cloud condensation nuclei artificially introduced into stratocumulus clouds, Atmospheric Chemistry and Physics, 15, 5645-5658, 2015.*
    - *Rosenfeld, D., Axisa, D., Woodley, W. L., and Lahav, R.: A quest for effective hygroscopic cloud seeding, Journal of Applied Meteorology and Climatology, 49, 1548-1562, 2010.*
    - *Ghate, V. P., Albrecht, B. A., Kollias, P., Jonsson, H. H., and Breed, D. W.: Cloud seeding as a technique for studying aerosol⊖cloud interactions in marine stratocumulus, Geophysical Research Letters, 34, 2007.*
    - *Wang, F., Li, Z., Jiang, Q., Wang, G., Jia, S., Duan, J., and Zhou, Y.: Evaluation of hygroscopic cloud seeding in liquid-water clouds: a feasibility study, Atmospheric Chemistry and Physics, 19, 14967-14977, 2019.*

- Response to Comment 1: We expanded the introduction of Section 5.3 to provide more context on aerosol-cloud interactions, particularly the role of giant CCN and their suggested concentrations for hygroscopic seeding.

- Changes to Manuscript:

    Line 321–348: "Ghate et al. (2007) studied the impact of introducing giant (salt) seeding aerosols (1–5 µm) into marine stratocumulus clouds using in situ aircraft observations off the central coast of

California. Seeding plumes were identified using a threshold of 250 cm$^{-3}$ for the PCASP concentrations compared to a background concentration of ~80 cm$^{-3}$. They observed a 5-fold increase in the number of large drops (20–40 μm) relative to the background, which was attributed to the activation of the seeding GCCN – a small fraction of the total aerosols produced by the flares. Furthermore, Jung et al. (2015) tested even larger seeding particles (1–10 μm) again in marine stratocumulus clouds off the central coast of California and reported a 4-fold increase in the rainfall rate associated with seeding GCCN concentrations of 10$^{-2}$–10$^{-4}$ cm$^{-3}$. More recently, Wang et al. (2019) reported on a cloud seeding case study over the eastern coast of Zhejiang, China and observed the hygroscopic growth of larger-mode seeding particles (>2 μm) up to a limit of ~18 μm drop sizes associated with the competition effect.

The characteristics of the background aerosol population, namely their size, concentration and chemical composition are considered key precursory properties to determine, and potentially improve, the effectiveness of seeding. Segal et al. (2004) report optimum seeding CCN concentrations of 700 cm$^{-3}$ in Mediterranean and extreme continental background conditions. This concentration is unrealistic in seeding operations and does not account for the impact of large background CCN which is further investigated by their simulations comparing seeded parcels with/without large, natural CCN centered on a diameter of 0.6 μm with concentrations of 0.15 and 0.3 cm$^{-3}$. Their results show a decrease in seeding impact when the large, natural CCN concentrations increased from 0.15 to 0.3 cm$^{-3}$. This was attributed to the competition with the prescribed seeding particles centered on a 10 μm diameter with a concentration of 0.032 cm$^{-3}$. Moreover, the original calculations of Ivanova et al. (1977) suggest CCN diameters larger than 5 μm serve as efficient raindrop embryos, while Segal et al. (2007) establish a minimum concentration of 0.025 cm$^{3}$ for such particles to cause a noticeable increase in warm rain production from a rising cloud parcel under typical conditions in Texas.

The UAE measurements show natural GCCN diameters (5–10 um) concentrations between 0.25–0.15 cm$^{-3}$ which are an order of magnitude larger than the seeding concentration suggested by Segal et al. (2004, 2007). Also, the UAE sub-cloud aerosol sizes extend from 0.01–100 μm with total concentrations ranging from 500–800 cm$^{-3}$. Hence, all three conceptual models for hygroscopic seeding outlined by Rosenfeld et al. (2010) are applicable to clouds studied over the UAE, namely, accelerating collision-coalescence by the competition effect (~1 μm), broadening the cloud drop size distribution by the tail effect (1–10 μm), and introducing ultra-giant seeding particles (>10 μm) to serve as rain drop embryos. These effects need to be thoroughly tested in model simulations based on the observations presented here."

Line 364–365: "The modelling work with different seeding material is in progress and is summarized in Geresdi et al. (2021)."

2) Comment 2: In section 5 It looks interesting that almost all the droplets in the negative temperature zone are supercooled water. According to glaciogenic seeding theory, does it mean the rich potential of cloud seeding in the UAE region?

- Response to Comment 2: The primary objective of the UAE campaign was carried out to investigate a potential secondary ice process (SIP) that may be activated by large amounts of super-cooled liquid drops in the subzero levels of mixed-phase clouds (Lawson et al., 2017). Our results indicate that the collision-coalescence (C-C) process was not activated in these clouds which suggests a low potential

for a natural SIP in upper levels. Modeling studies can help assess the effectiveness of perhaps larger hygroscopic seeding particle sizes (10–15 μm), relative to background aerosols, in initiating C-C a potential SIP.

In terms of glaciogenic seeding potential, as noted by Kumar and Suzuki (2019), the large amounts of super-cooled liquid water observed in clouds over the northeastern UAE, especially during the winter season, may be transformed into ice by the ingestion of ice nuclei. Further modeling studies incorporating the in situ observations in this paper can help assess the potential of galciogenic seeding for UAE clouds, where current operations are limited to hygroscopic seeding at cloud base.

- Changes to Manuscript: Lines 384–388: "Furthermore, no indication of C-C is observed within any of the upper levels listed in Table 2 and displayed in Figures 9, 10 and 11. In the upper levels of SF1 (-12.6 and -12.4 °C), a dominant population of liquid drops (d<50 μm) is observed with very few ice particles showing a habit of sector plates (expected by nucleation at -12 °C). LWCs of ~1.4 $g.m^{-3}$ with strong updrafts (~17.8 $m.s^{-1}$) and MVDs less than 20 μm are observed at these sub-freezing levels. Similar observations are also recorded in the upper levels of SF4 with no signs of ice multiplication."

3) Comment 3: Line 255-258: Please show the relationship between vertical velocity and spatial position (or time series) during cloud penetration with a diagram to illustrate the huge difference of updraft (17.8 m $s^{-1}$) and downdraft (-12.4 m $s^{-1}$) measured in the upper portion of SF1.

- Response to Comment 3: The below figure shows the variation of the vertical velocity during the first two penetrations in the upper-level of SF1.

- Changes to Manuscript: The below figure is added as an inset in Figure 7a and referenced on Page 25.

[Figure]

4) Comment 4: Line 266-267: Why does drop size in the lower portion of SF1 seem smaller than that of SF4 from CIP image? As the fallout of ice irregulars or graupel are observed in both cloud penetrations.

- Response to Comment 4: Very few ice particles with a habit of sector plates are captured by the CPI at -12.4 °C in a decaying turret from SF1 (see Figure 9b and tail of Figure 11a). Alternatively, a relatively larger number of mm-sized irregulars and graupel are observed at -10.6 °C in a growing turret from

SF4 (see Figure 10b and tail of Figure 11b). This explains the larger contribution from fallout ice to the lower portions of SF4 compared to SF1.

5) Comment 5: Line 274-276: In contrast to 8.3 °C and -0.3 °C, 8.6 °C and 8.3 °C are almost at the same height during SF4 cloud penetration, please explain why spectrum broadening is obviously observed.

- Response to Comment 5: Despite the marginal altitude difference (~120 m) between the 8.6 °C and 8.3 °C levels, Figure 10 shows that their penetrations are borderline – just below cloud base and within the sub-cloud region, respectively. Broadening is therefore more pronounced at the 8.3 °C level, which transitions from out-of-cloud to in-cloud conditions.

6) Comment 6: Figure 9 and 10: How to determine the red oval in camera photo corresponds to the measurement by 2ds and cpi? Please add descriptions.

- Response to Comment 6: The cloud penetration locations (red ovals in Figures 9/10) were determined by visually inspecting video footage from the forward-facing cockpit camera within 1 minute of each cloud approach. A description is added to the caption on Figure 9 (Page 27).

**Technical corrections**

1) Line 93: "in favour of" -> "in favor of".

   Line 93: corrected.

2) Line 161: "compliment" -> "complement".

   Line 161: corrected.

3) Line 205-206: "…with high concentrations of around 1000 cm$^{-3}$…" -> "…with high concentrations of aerosols (around 1000cm$^{-3}$) …".

   Line 205-206: revised to "…with high concentrations of aerosols (~ 1000cm$^{-3}$) …".

4) Line 254: "-12 C" -> "-12 °C".

   Line 267: corrected.

5) Line 287: What is "PSD" short for? Please give the full name of the acronym when it first appears.

   Line 300: "… particle size distribution (PSD) …" – the spell out of all acronyms at first use was checked.

6) Figure 1: Please mark the location of the airport.

   Page 19: The location of the airport is marked as "Al Ain" on Figure 1 and stated in the figure caption.

7) Figure 2 and 3: Please improve the graph resolution.

The quality and resolution of Figures 2 and 3 have been improved.

8) Table1: The second annotation was not marked on the table.

Corrected.

9) Table 2: Please add standard deviation of the data.

Standard deviations added in Table 2.

10) Reference: Please unify the format of journal titles, such as "Atmospheric Chemistry & Physics" and "Atmospheric Chemistry and Physics, "Atmospheric environment" and "Atmospheric Environment" …

Journal titles are now unified in the reference list.

**Authors' References**

- Geresdi, I., Chen, S., Wehbe, Y., Bruintjes, R., Lee, J., Tessendorf, S., Weeks, C., Sarkadi, N., Rasmussen, R. M., and Grabowski, W.: Sensitivity of the Efficiency of Hygroscopic Seeding on the Size Distribution and Chemical Composition of the Seeding Material, 101st American Meteorological Society Annual Meeting, 2021,
- Ghate, V. P., Albrecht, B. A., Kollias, P., Jonsson, H. H., and Breed, D. W.: Cloud seeding as a technique for studying aerosol-cloud interactions in marine stratocumulus, Geophysical research letters, 34, 2007.
- Ivanova, E., Kogan, Y., Mazin, I., and Permyakov, M.: The ways of parameterization of condensation drop growth in numerical models, Izv. Atmos. Oceanic Phys, 13, 1193-1201, 1977.
- Jung, E., Albrecht, B. A., Jonsson, H. H., Chen, Y.-C., Seinfeld, J. H., Sorooshian, A., Metcalf, A. R., Song, S., Fang, M., and Russell, L. M.: Precipitation effects of giant cloud condensation nuclei artificially introduced into stratocumulus clouds, Atmospheric Chemistry and Physics, 15, 5645-5658, 2015.
- Kumar, K. N., and Suzuki, K.: Assessment of seasonal cloud properties in the United Arab Emirates and adjoining regions from geostationary satellite data, Remote sensing of environment, 228, 90-104, 2019.
- Lawson, P., Gurganus, C., Woods, S., and Bruintjes, R.: Aircraft observations of cumulus microphysics ranging from the tropics to midlatitudes: Implications for a "new" secondary ice process, Journal of Atmospheric Sciences, 74, 2899-2920, 2017.
- Rosenfeld, D., Axisa, D., Woodley, W. L., and Lahav, R.: A quest for effective hygroscopic cloud seeding, Journal of applied meteorology and climatology, 49, 1548-1562, 2010.
- Segal, Y., Khain, A., Pinsky, M., and Rosenfeld, D.: Effects of hygroscopic seeding on raindrop formation as seen from simulations using a 2000-bin spectral cloud parcel model, Atmospheric Research, 71, 3-34, 2004.
- Segal, Y., Pinsky, M., and Khain, A.: The role of competition effect in the raindrop formation, Atmospheric research, 83, 106-118, 2007.
- Wang, F., Li, Z., Jiang, Q., Wang, G., Jia, S., Duan, J., and Zhou, Y.: Evaluation of hygroscopic cloud seeding in liquid-water clouds: a feasibility study, Atmospheric Chemistry and Physics, 19, 14967-14977, 2019.